# A durable and pH-universal self-standing MoC–Mo$_2$C heterojunction electrode for efficient hydrogen evolution reaction

Wei Liu [1,2], Xiting Wang [3], Fan Wang[1,2], Kaifa Du[1,2], Zhaofu Zhang [4], Yuzheng Guo [3✉], Huayi Yin[1,2✉] & Dihua Wang [1,2,5✉]

Efficient water electrolyzers are constrained by the lack of low-cost and earth-abundant hydrogen evolution reaction (HER) catalysts that can operate at industry-level conditions and be prepared with a facile process. Here we report a self-standing MoC–Mo$_2$C catalytic electrode prepared via a one-step electro-carbiding approach using CO$_2$ as the feedstock. The outstanding HER performances of the MoC–Mo$_2$C electrode with low overpotentials at 500 mA cm$^{-2}$ in both acidic (256 mV) and alkaline electrolytes (292 mV), long-lasting lifetime of over 2400 h (100 d), and high-temperature performance (70 °C) are due to the self-standing hydrophilic porous surface, intrinsic mechanical strength and self-grown MoC (001)–Mo$_2$C (101) heterojunctions that have a $\Delta G_{H*}$ value of −0.13 eV in acidic condition, and the energy barrier of 1.15 eV for water dissociation in alkaline solution. The preparation of a large electrode (3 cm × 11.5 cm) demonstrates the possibility of scaling up this process to prepare various carbide electrodes with rationally designed structures, tunable compositions, and favorable properties.

[1] School of Resource and Environmental Science, Wuhan University, Wuhan 430072, China. [2] International Cooperation Base for Sustainable Utilization of Resources and Energy in Hubei Province, Wuhan University, Wuhan 430072, China. [3] School of Electrical Engineering and Automation, Wuhan University, Wuhan 430072, China. [4] Department of Engineering, University of Cambridge, Cambridge CB2 1PZ, UK. [5] State Key Laboratory of Water Resources and Hydropower Engineering Science, Wuhan University, Wuhan 430072, China. ✉email: yguo@whu.edu.cn; yinhuayi@whu.edu.cn; wangdh@whu.edu.cn

Hydrogen ($H_2$) has long been considered as a clean energy carrier that can be obtained from unlimited water and renewable power, and the annual demand for clean $H_2$ is estimated to exceed 75 M tons by 2030[1–3]. High-efficient, low-cost, Earth-abundant, and long-lasting lifetime HER catalysts are a key enabler for water electrolysis that produces $H_2$ with minimum energy consumption and cost[4–6]. Pt-based electrocatalysts have shown superior catalytic performance with low overpotential, fast charge transfer kinetics, and long durability[7,8]. Unfortunately, the Pt-based catalysts cannot meet the ever-increasing demand in water electrolyzers and fuels cells because of the limited resources and poor activity in alkaline solution[9,10]. Among various non-precious metal catalysts, molybdenum carbides (MoC, $Mo_2C$) have emerged as a promising HER catalyst due to their similar d-band electronic structure to Pt[11,12]. However, the HER catalytic activities of MoC or $Mo_2C$ have not reached expected performances that are close to Pt[13,14]. Element doping or material hybridization is a common way to improve the catalytic activity of MoC and $Mo_2C$ by tailoring their band structure and thereby creating more active sites for HER[15–17]. Recently, the MoC–$Mo_2C$ composite powdery catalyst showed improved electrocatalytic activity[18,19]. However, the underlying mechanism remains unclear. In addition, the powdery molybdenum carbide composite catalyst operates at a relatively low current density (20 mA $cm^{-2}$) for less than 30 h at room temperature, which is not able to meet the requirements of industrial applications. Therefore, it is important to explore novel methods to prepare molybdenum carbides with enhanced robustness, high catalytic activity, long-lasting lifetime, and superior stability at high current densities (400–600 mA $cm^{-2}$) and high operating temperatures (~70 °C) like the industrial water electrolyzer.

Electrode design and surface engineering are effective ways to improve the HER performance such as the enhanced tolerance at a wide $pH$ range, high current densities, and high operating temperatures. For example, a self-standing $MoS_2/Mo_2C$ electrode grown on a titanium (Ti) foil by a chemical vapor deposition method (CVD) showed excellent HER performance at 1000 mA $cm^{-2}$ and a long lifetime of over 1000 h in both acidic and alkaline solutions[20]. Likewise, many other approaches can be applied for preparing MoC/$Mo_2C$-based self-standing HER catalytic layer on a conductive substrate, such as Versatile Electropolymerization-Assisted Method[21], Template Directed Strategy[18], Wet Chemistry Synthesis[22], and Sublimation-Reduction Method[23]. Although many of $Mo_xC$-based self-standing electrodes prepared by the above methods exhibit relatively enhanced HER performance, these preparation processes involve multiple steps, which are energy-intensive and increase progress complexity. More importantly, most synthesized catalyst layers cannot withstand a high current density and operate for a long-lasting lifetime because of the weak interfacial adhesion on the electrode substrate, and a robust self-standing MoC–$Mo_2C$ composite catalyst layer has not been achieved yet.

In this work, we prepared a self-standing MoC–$Mo_2C$ hetero-junction catalytic layer on a Mo sheet in molten carbonate where $CO_2$ was electrochemically reduced to C that simultaneously reacted with the Mo substrate to form a porous MoC–$Mo_2C$ heterojunction catalytic layer. Unlike conventional approaches using an inert substrate, the molten carbonate electrolysis employs an active substrate (e.g., Mo) that is involved in the formation of the carbide layer, thereby ensuring the firm connection between the Mo substrate and the electrolytic MoC–$Mo_2C$ heterojunction layer. Thus, the one-step prepared electrolytic MoC–$Mo_2C$ heterojunction layer is robust enough to survive at a high current density of 500 mA $cm^{-2}$, a long-lasting lifetime of over 2400 h, in both acidic (256 mA, 62 mV $dec^{-1}$) and alkaline (292 mA, 59 mV $dec^{-1}$) solutions, and at the industrial operating temperature of

~70 °C. In addition, the underlying mechanism of the enhanced HER performance of the MoC–$Mo_2C$ heterojunction is revealed by electronic structure calculations based on density functional theory (DFT). Lastly, the molten carbonate electrolysis is proven to be an effective surface engineering approach to tailoring the hydrophily of the surface (contact angle is about 32°), the composition of the catalytic layer, as well as the surface micro-topography. Overall, both experimental and theoretical studies confirm that the electrolytic MoC–$Mo_2C$ heterojunction is a promising low-cost catalyst for $H_2$ production operating at simulated industrial operating conditions.

## Results

**Synthesis of MoC–$Mo_2C$ heterojunctions**. A carbide layer can be electrochemically constructed at the surface of a Mo cathode coupled with an oxygen-evolution inert anode (Fig. 1a). Unlike the conventional electrodeposition process, the Mo electrode substrate takes part in the electrodeposition process that contains two steps. The first step is the electrochemical reduction of $CO_3^{2-}$ at the cathode, generating carbon while releasing $O^{2-}$ that is partly consumed by the carbonization of $CO_2$ to replenish $CO_3^{2-}$ and partly discharging at the anode to generate $O_2$. The second step is the carbiding reaction between the electrolytic carbon and the Mo substrate. Thus, the so-called electro-carbiding reaction employs the $CO_2$ as the feedstocks and electrons as the reducing agent, laying a sustainable way to utilize $CO_2$ as a valuable feedstock. After electrodeposition at 790 °C for 2 h, the Mo electrode with a metallic luster turned to gray, suggesting that a uniform layer had been deposited on the surface of the Mo electrode. The gray deposit is in the form of interconnected honeycomb-like porous architectures (Fig. 1b), and the diameter of these interconnected pores ranges from 50 to 200 nm (the inset of Fig. 1b). The porous structure of MoC-$Mo_2C$-790 is further confirmed by the TEM image (Fig. 1c). The XRD patterns show that the deposit consists of a mixture of MoC (PDF#65-3494) and $Mo_2C$ (PDF#35-0787) (Fig. 1g), and distinct interfaces can be seen between MoC (001) and $Mo_2C$ (101) (Fig. 1d). In addition, the corresponding mapping results from TEM and EPMA tests exhibit the uniform distribution of C and Mo in the as-prepared $Mo_xC$ films (Fig. 1e). XPS results in Fig. 1h show that $Mo^{2+}$ and $Mo^{3+}$ originate from MoC and $Mo_2C$[24,25], and $Mo^{4+}$ and $Mo^{6+}$ species derive from $MoO_2$ and $MoO_3$ that are often observed at the surface of MoC/$Mo_2C$ when exposed to water or air[26–28]. Supplementary Fig. 1a shows the high-resolution spectrum in the C 1s region that was fitted by components corresponding to Mo–C, C–C, C–O, and C=O species[15,29]. The peak at a BE of 282.8 eV is assigned to the Mo-C species and represent 12 at% of the C 1s region. It should be noted that the 12 at% of the C1s region does not mean that the amount of Mo-C is 12 at%, but rather the fact that thin carbon layers formed on the surface of the electrode. The O 1s XPS spectrum (Supplementary Fig. 1b) indicates the slight oxidation of the Mo-based composites, which is in consistent with the Mo 3d results in Fig. 1h. Thus, the molten carbonate electrolysis is an effective method to prepare a porous carbide layer containing MoC-$Mo_2C$ heterojunctions.

The electrolysis temperature significantly affects the composition of carbide films. As shown in Supplementary Fig. 2, the electrode prepared at 590 °C still maintained the metallic luster, meaning that Mo was inert during the electrolysis at this temperature. The electrodes obtained at 690 °C and 890 °C were covered by a layer that was generated by the electro-carbiding. From SEM images, the surfaces of electrodes obtained at different temperatures show different microstructures (Supplementary Fig. 3). More importantly, the composition of the electrolytic film varies with altering the electrolysis temperature. The composition of the electrode surface is Mo at 590 °C, and a mixture of MoC and $Mo_2C$ exists at the

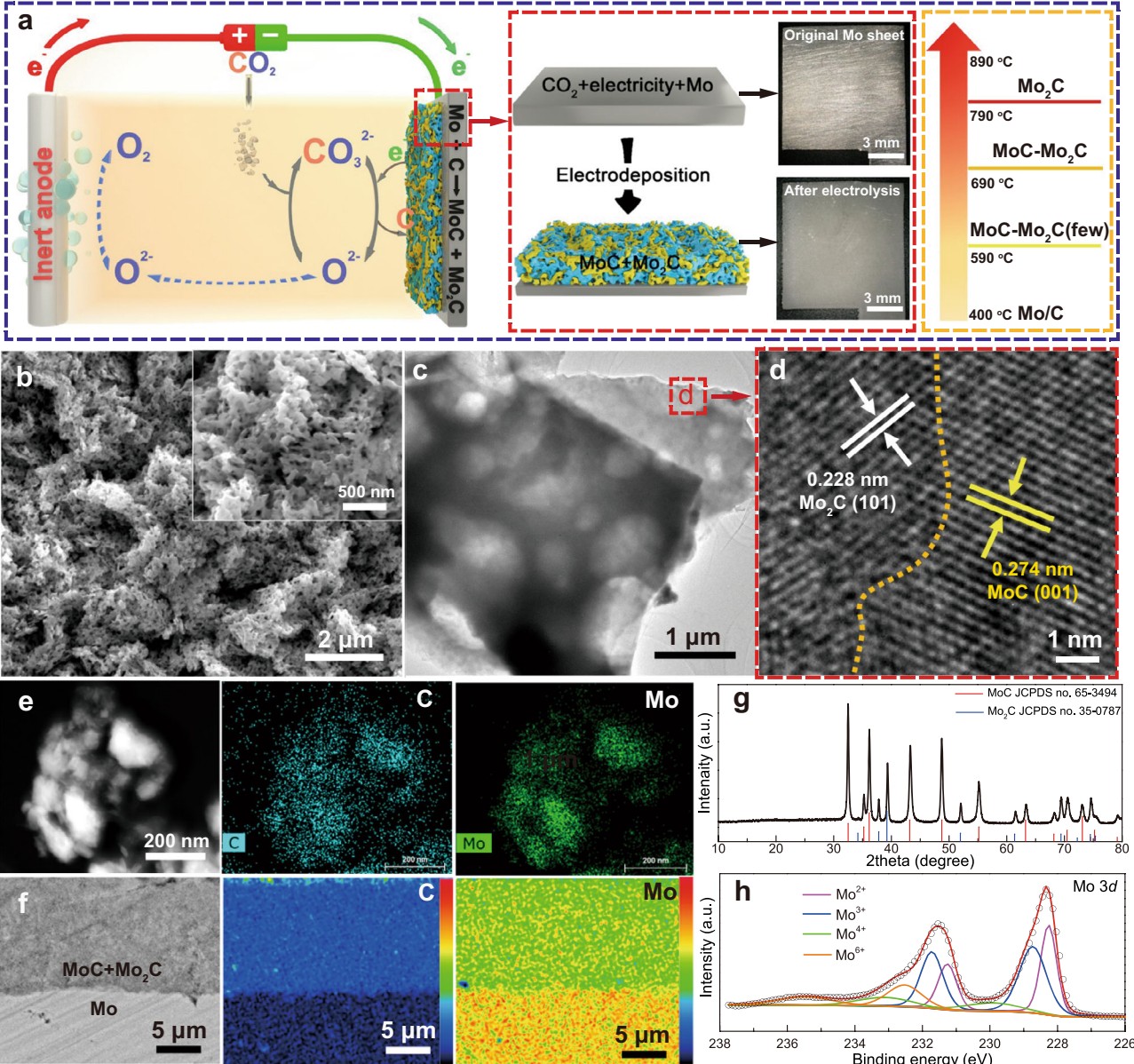

**Fig. 1 Materials synthesis and characterization. a** Schematic of preparing the MoC–Mo$_2$C-790 in molten carbonate and digital pictures of the Mo electrode before and after electrolysis. **b** SEM images of the synthesized MoC–Mo$_2$C-790, the inset is an enlarged view of the 3D hierarchical honeycomb-like structures on the surface of MoC–Mo$_2$C-790. **c** TEM image of the samples collected from the MoC–Mo$_2$C-790. **d** High-resolution transmission electron microscopy (HRTEM) image of the powders from the MoC–Mo$_2$C-790. **e** HAADF-STEM image and the corresponding EDS mapping of the MoC–Mo$_2$C-790. **f** Cross-sectional SEM images (BSE) with their corresponding element distribution mappings by EPMA of the MoC–Mo$_2$C-790. **g** XRD pattern of the MoC–Mo$_2$C-790. **h** XPS pattern of the MoC–Mo$_2$C-790.

surface of the electrode and MoC is the dominating phase at 690 °C. The intensity of Mo$_2$C increases in the products obtained at 790 °C, and only Mo$_2$C exists in the electrode obtained at 890 °C (Supplementary Fig. 4). It should be noted that carbon is uniformly distributed in the surface layer, but the Mo element shows distinct different distributions in different areas in the EPMA test (Fig. 1f). The different Mo distributions are due to the different Mo contents of MoC and Mo$_2$C. Meanwhile, the MoC–Mo$_2$C-690 electrode exhibits an obvious stratification (Supplementary Fig. 5a–c, the upper layer is MoC and the lower layer is Mo$_2$C). Carbon and Mo distributes uniformly at the Mo$_2$C-890 electrode (Supplementary Fig. 5d–f), which agrees well with XRD results. Thus, an operating temperature higher than 590 °C can initiate the carbiding reaction and the MoC prefers to form at a low temperature and Mo$_2$C is the favorable product when the temperature is higher than 890 °C.

And the valence bonds of the surface vary with the compositions of the electrodes obtained at different temperatures, further confirming the composition and chemistry of the electrode surface can be tailored by adjusting the electrolysis temperature (Supplementary Fig. 6). Therefore, the compositions and structures of the carbide film can be well modulated by tuning the electrolysis temperature that governs the diffusion kinetics of carbon and molybdenum at the interface of Mo/carbide layer.

**HER performances**. The HER performances of electrolytic self-standing electrodes were evaluated in both acidic and alkaline solutions. As shown in Fig. 2a and Supplementary Fig. 7a, the MoC–Mo$_2$C-790 electrode only has a small overpotential of 114 mV and 183 mV to reach a geometric current density ($j$) of 10 mA cm$^{-2}$ and 100 mA cm$^{-2}$ in H$_2$SO$_4$ solution, which are

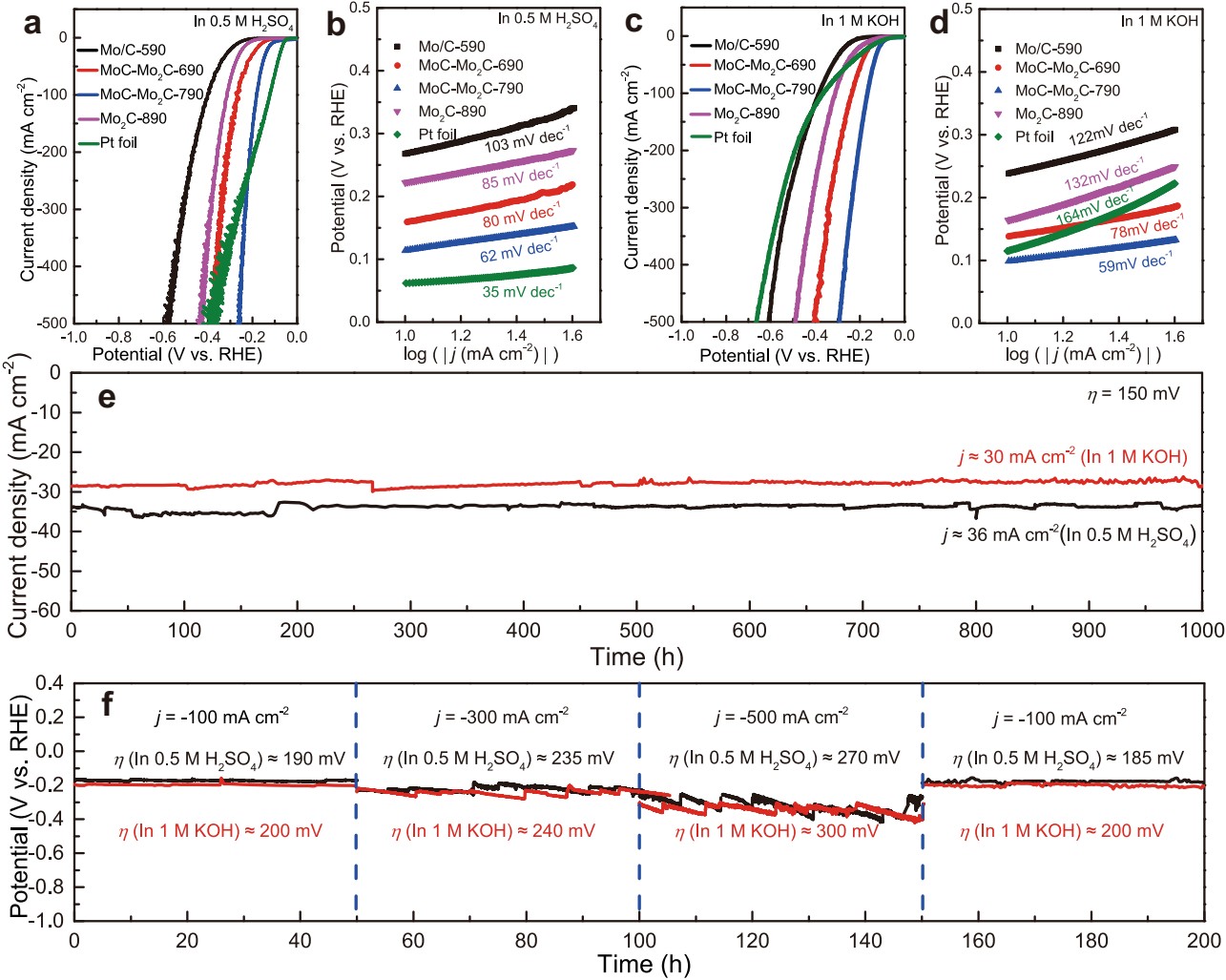

**Fig. 2 HER performances of electrolytic electrodes in both acidic and alkaline solutions. a** Polarization curves (with *iR*-correction) and **b** their corresponding Tafel plots of various electrodes in 0.5 M H$_2$SO$_4$. **c** Polarization curves (with *iR*-correction) and **d** corresponding Tafel plots of electrolytic electrodes in 1 M KOH. **e** The long-term electrolysis stability test of MoC–Mo$_2$C-790 at $\eta$ = 150 mV in 0.5 M H$_2$SO$_4$ and 1 M KOH. **f** The stability test at 100 mA cm$^{-2}$, 300 mA cm$^{-2}$ and 500 mA cm$^{-2}$ in 0.5 M H$_2$SO$_4$ (in black) and 1 M KOH (in red).

much lower than that of Mo/C-590, MoC–Mo$_2$C-690 and Mo$_2$C-890 electrodes (for details see Supplementary Table 1). Moreover, the overpotential of the MoC-Mo$_2$C-790 electrode ($\eta$ = 256 mV) is smaller than Pt ($\eta$ = 410 mV) at a high current density of 500 mA cm$^{-2}$ and is better than most HER catalysts that have been reported (Supplementary Table 2). Accordingly, the derived Tafel plots from Fig. 2a indicate that the MoC-Mo$_2$C-790 electrode has the smallest Tafel slope value of 62 mV dec$^{-1}$ among all electrolytic electrodes (Fig. 2b). Although the Tafel slope value of the MoC-Mo$_2$C-790 electrode is slightly higher than that of Pt (35 mV dec$^{-1}$), the slope value of the MoC–Mo$_2$C-790 (110 mV dec$^{-1}$) electrode at the strong polarization zone (at 200 to 500 mA cm$^{-2}$) is much smaller than that of Pt (405 mV dec$^{-1}$) (Supplementary Fig. 8). This means that the MoC–Mo$_2$C-790 electrode is able to operate at high current densities with a smaller overpotential, which could be an indicator to evaluate the performance of a catalyst at large current densities and is meaningful for practical use. The Nyquist plots also confirm that the MoC–Mo$_2$C-790 electrode has the smaller charge transfer resistance than other electrodes at various applied overpotentials (Supplementary Fig. 9). At $\eta$ = 130 mV, the MoC-Mo$_2$C-790 electrode shows a much lower $R_{ct}$ value (2.69 Ω) in comparison with other electrolytic electrodes (107.6 Ω for

Mo/C-590, 7.09 Ω for MoC-Mo$_2$C-690 and 23.26 Ω for Mo$_2$C-890) (Supplementary Figs. 10 and 11, Table 1), further demonstrating its fast charge-transfer rate. Besides in acidic solution, the HER performance of the MoC–Mo$_2$C-790 electrode was also evaluated in alkaline solution. The electrode affords a current density 10 mA cm$^{-2}$ and 500 mA cm$^{-2}$ at small overpotentials of 98.2 mV and 292 mV with a Tafel slope value of 59 mV dec$^{-1}$ in 1 KOH solution (Fig. 2c, d, Supplementary Fig. 7b), verifying the outstanding activity among most of reported self-standing noble metal-free HER electrodes (Supplementary Table 3). Note that similar Tafel slope values in acidic and alkaline solutions indicate that the acid- and base-tolerance of the MoC–Mo$_2$C composite electrode surpasses precious metal catalysts (Pt), especially in alkaline solution (Fig. 2b, d).

In addition to its excellent HER activity in both acidic and alkaline solutions, the MoC-Mo$_2$C-790 electrode also exhibits super electrocatalytic stability. As shown in Fig. 2e, the MoC–Mo$_2$C-790 electrode shows an almost constant catalytic current for 1000 h (2000 h in total for the same electrode) at $\eta$ = 150 mV in 0.5 M H$_2$SO$_4$ and 1 M KOH, respectively. Furthermore, the high-current-density performances of the electrode were evaluated at 100, 300, and 500 mA cm$^{-2}$ in 0.5 M H$_2$SO$_4$ and 1 M KOH for over 400 h, respectively (Fig. 2f).

After operating at high current densities, the overpotential returned back to its original value, demonstrating its excellent stability even at an industrial-level current density. Notably, the MoC-Mo$_2$C-790 electrode maintains the HER activity after operating for 2400 h (Supplementary Fig. 12). Thus, the MoC-Mo$_2$C-790 electrode is proved to be an ultra-stable electrode for HER at both acidic and alkaline solutions, which outperforms most of reported HER catalysts (Supplementary Tables 4 and 5). After 2400 h's stability test, the composition of the catalytic layer does not change (Supplementary Fig. 13), and the Mo 3d spectrum remains the same as the new electrode (Supplementary Fig. 14). Besides, the surface of the MoC-Mo$_2$C-790 electrode retains its origins 3D hierarchical honeycomb-like nanostructures and intact MoC and Mo$_2$C lattice structure (Supplementary Figs. 15 and 16). Unlike some Pt-based catalysts that may have cathodic corrosion[30], the structural and chemical stability of the MoC-Mo$_2$C heterojunctions ensures the super stability even under a high current density for a long service time. Moreover, the 17 N bond strength between the MoC-Mo$_2$C coating and the Mo substrate (Supplementary Fig. 17) and the unchanged thickness of the MoC-Mo$_2$C coating before and after the stability test (Supplementary Fig. 18) verify that the electrolytic catalytic layer can withstand the erosion of H$_2$ gases for a long time. The firm adhesion with the Mo substrate stems from the use of an active Mo electrode substrate that takes part in the formation of the catalytic layer. Hence, the MoC-Mo$_2$C coating not only acts as the HER catalytic layer but also a corrosion barrier to protect the substrate in both sulfuric acid and potassium hydroxide solutions (Supplementary Fig. 19).

**Mechanism and theoretical calculation.** In the HRTEM images (Fig. 1d), an interfacial heterostructure can be evidently observed between the MoC and Mo$_2$C nanodomains, as displayed by dashed lines. This kind of interfacial nanodomains was also observed in other multicomponent materials such as WO$_x$/WC[31], Mo$_2$N/Mo$_2$C[15], MoS$_2$/Mo$_2$C[20] and NiCo$_2$S$_4$/Ni$_3$S$_2$[32]. And these heterojunction interfaces in the above-mentioned materials were proved to contain some highly electrocatalytic active sites. Figure 3a shows the TOF (calculation details are given in ESI) of various HER electrodes. At an overpotential of 250 mV, the TOF value of the MoC–Mo$_2$C-790 electrode is calculated to be 1.3 s$^{-1}$, which is dramatically larger than the other electrodes, indicating the excellent intrinsic activity of the MoC-Mo$_2$C-790 electrode. It is known that the HER activity of Mo$_x$C is closely dependent on the active Mo$^{2+}$ and Mo$^{3+}$ species, which presents different Mo–H bonds because of their various electron densities[33,34]. As shown in Supplementary Fig. 20 and Supplementary Table 6, different HER electrodes with different contents of MoC or Mo$_2$C have different Mo$^{3+}$/Mo$^{2+}$ mole ratios (n$_{3+/2+}$). Supplementary Fig. 20 indicates that the HER performance of the four electrodes depends on the variation of Mo$^{3+}$/Mo$^{2+}$ mole ratios (n$_{3+/2+}$) on the electrode surface, which is reflected by the exchange current density (j$_0$). The MoC–Mo$_2$C-790 electrode presents an apparently improved HER activity because it has a higher Mo$^{3+}$/Mo$^{2+}$ mole ratio (n$_{3+/2+}$ = 1.42) than that of the Mo$_2$C-890 electrode (n$_{3+/2+}$ = 0.61), indicating that the added Mo$^{3+}$ species are beneficial for HER. According to XRD and XPS patterns, the composition of the film on Mo$_2$C-890 was pure Mo$_2$C with the dominance of Mo$^{2+}$. The strong Mo-H on Mo$_2$C benefits H$^+$ reduction (i.e., the Volmer step), but limits the desorption of adsorbed H (H$_{ads}$) (i.e., the Heyrovsky step)[35,36]. The reduction of electron density in the MoC–Mo$_2$C-790 electrode for the formation of MoC-Mo$_2$C heterojunction with the ascendency of Mo$^{3+}$ weakens the strength of Mo–H towards accelerating H$_{ads}$ desorption, and thus significantly enhances the HER performance.

With increasing n$_{3+/2+}$ to 1.98, the MoC–Mo$_2$C-690 electrode with a main MoC phase shows an obviously decreased HER activity comparing with the MoC–Mo$_2$C-790 electrode.

The DFT calculations on both superficial and interfacial Mo sites of MoC, Mo$_2$C and MoC–Mo$_2$C were conducted to compare the electrocatalytic HER activity of bare MoC, Mo$_2$C and MoC–Mo$_2$C heterojunction. On acidic condition, among MoC (001), Mo$_2$C (101), and the MoC–Mo$_2$C, the MoC–Mo$_2$C heterojunction exhibits the optimum Gibbs free energy of H* adsorption ($\Delta G_{H*} = -0.13$ eV), as shown in Fig. 3b. HER electrocatalyst with a positive value result in the poor adsorption of H*, while a catalyst with a negative value may lead to the difficult release of a H$_2$. The ideal value of $|\Delta G_{H*}|$ should be zero. The smallest value of $|\Delta G_{H*}|$ of the MoC–Mo$_2$C heterojunction indicates its better activity. The models of H* adsorption for Mo$_2$C (101) and MoC (001) are compared in Supplementary Fig. 21. The lattice mismatch of MoC and Mo$_2$C causes large local distortion at the heterostructures and changed the local electronic structure with more Mo$^{3+}$/Mo$^{2+}$ sites, consistent with our previous experiments. Bader charge analysis was conducted to analyze the charge transfer[37]. H adatom adsorbed at the MoC–Mo$_2$C heterojunction possesses the least charge transfer of 0.35 e, corresponding to the smallest adsorption energy of −0.28 eV. There is a linear relationship between the charge transfer and the adsorption energy $E_{ads}$ of the H adatom with a quite well correlation coefficient, $R^2 = 0.989$, as shown in Supplementary Fig. 22. This indicates that more charge transfer on the H adatom leads to stronger adsorption of the H adatom. The charge density difference diagram also proves that the electron drifts from the substrate to the H atom (Supplementary Fig. 23). The projected crystal orbital Hamilton population (pCOHP) was introduced to analyze the contribution of bonding and antibonding states of H absorbed at the MoC–Mo$_2$C heterojunction interface[38,39] (Supplementary Fig. 24). The integrated COHP (ICHOP) was calculated by integrating the energy up to the Fermi level for quantitative bonding contribution. ICOHP is −0.94 for the MoC–Mo$_2$C, corresponding to moderate bonding strength. As can be seen in Supplementary Fig. 25, the overlapping peaks by H partial density of states are more noticeable and more closely to the Fermi level on the MoC–Mo$_2$C than pure MoC (001) or Mo$_2$C (101), indicating that the MoC–Mo$_2$C is beneficial to activate hydrogen atoms.

During the process of HER in alkaline solution, the first key step is that the first H$_2$O adsorbs on the surface and dissociates into intermediate H* and OH*. Then, the dissociation of the second H$_2$O leads to the generation of H$_2$ (Supplementary Figs. 26–28). The free energy diagrams are plotted to explain the corresponding HER performance in alkaline solution. As shown in Fig. 3c, the highest energy barrier (the second H$_2$O dissociation step) of the MoC-Mo$_2$C heterojunction is 1.15 eV, while the highest energy barrier on bare MoC and Mo$_2$C are 1.90 and 3.33 eV, respectively. This indicates that the energy barrier of water dissociation on the Mo$_2$C–MoC interface is lower than that on Mo$_2$C or MoC, which can be explained by the interface Mo $d$ orbital tuning since engineering the transition metal $d$ orbitals is a feasible method to modulate the interaction between the molecule and active sites[40,41]. The occupied orbitals of the H$_2$O molecule are mainly $p$ states. And the partial DOS shows that the electron transfer is mainly from the d orbitals of Mo atoms (Supplementary Fig. 29). The empty orbitals of MoC and Mo$_2$C near the Fermi level are more localized within the surface (Supplementary Fig. 30), suggesting that both MoC and Mo$_2$C are easy to capture the H$_2$O molecule but hard to dissociate the H$_2$O molecule during the adsorption. While the active Mo atoms on the MoC–Mo$_2$C interface have a higher partial DOS near Fermi level (Supplementary Fig. 29) and an empty hybridized $d$ orbital perpendicular to the surface, which is not only beneficial to capture the H$_2$O

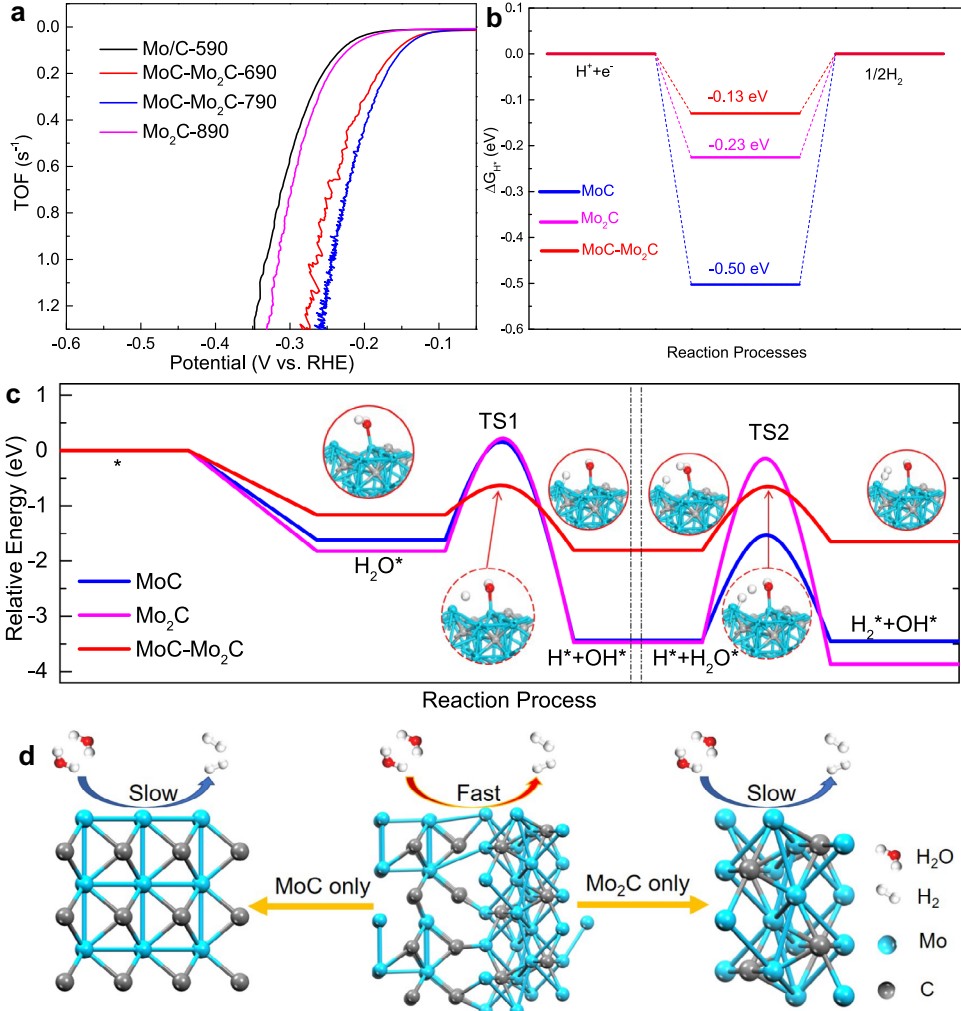

**Fig. 3 TOF LSV curves and DFT calculations. a** TOF LSV curves of different electrodes. **b** Calculated $\Delta G_{H^*}$ diagram of the HER in acid electrolyte at the equilibrium potential. **c** Relative energy diagram of water dissociation on MoC, $Mo_2C$, and MoC-$Mo_2C$, including the two steps of water dissociation, in alkaline solution, TS: Transition State. **d** Schematic illustration of the HER mechanism.

molecule but also easy to dissociate the $H_2O$. In addition, the electron overlap between the active site and $H_2O$ molecule on the MoC–$Mo_2C$ interface is moderate compared to MoC and $Mo_2C$, which is helpful to decrease the energy barrier (Supplementary Fig. 31). Overall, the energy barrier of water dissociation on the MoC–$Mo_2C$ interface is lower because of the Mo $d$ orbital tuning. Therefore, the MoC–$Mo_2C$ interface exhibits better HER activity than MoC and $Mo_2C$ in both acid and alkaline solutions (Fig. 3d).

**Surface properties**. The surface property of the electrode is paramount to determine the HER performance, especially at high current densities. First, the electrochemical active surface area (ECSA) of all HER electrodes were obtained by the double-layer capacitances ($C_{dl}$) that were derived from CV curves. The rectangular shape CV curves imply the high electrical conductivity of these electrodes (Supplementary Fig. 32), and the $C_{dl}$ values obtained from CVs are 5.68 mF cm$^{-2}$ for the Mo/C-590 electrode, 28.25 mF cm$^{-2}$ for the MoC–$Mo_2C$-690 electrode, 111.12 mF cm$^{-2}$ for the MoC–$Mo_2C$-790 electrode, and 34.12 mF cm$^{-2}$ for the $Mo_2C$-890 electrode (Supplementary Fig. 33). Accordingly, ECSA are 142 cm2 for Mo/C-590, 706 cm$^2$ for MoC–$Mo_2C$-690, 2778 cm$^2$ for MoC–$Mo_2C$-790, and 853 cm$^2$ for the $Mo_2C$-890 (Supplementary Table 7). Thus, the MoC–$Mo_2C$-790 electrode has the largest ECSA among the four electrodes, which

is beneficial for the HER catalytic performance. However, the HER performance is not only correlate to ECSA. For example, the HER catalytic performance of the $Mo_2C$-890 electrode is obviously lower than that of the MoC–$Mo_2C$-690 electrode although the ECSA of the $Mo_2C$-890 electrode (853 cm$^2$) is larger than the ECSA of the MoC–$Mo_2C$-690 electrode (706 cm$^2$) Thus, the HER performance is mainly determined by both the composition and the active surface area of the electrode.

In addition to the surface area, the hydrophilicity of the electrode surface affects $H_2$ bubbles' motion behaviors that determines the kinetics of the HER performance. In principle, it's better to form uniform and small $H_2$ bubbles that can leave from the electrode without causing high polarization of the electrode[34,42]. The contact angles of Mo/C-590, MoC–$Mo_2C$-690, MoC–$Mo_2C$-790 and $Mo_2C$-890 electrodes are 76º, 68º, 32º, and 54º (Fig. 4), respectively. The best wettability of the MoC–$Mo_2C$-790 electrode may be attributed to its 3D hierarchical honeycomb-like nano-porous architectures with open channels/pores, which is expected to benefit the bubble separation and allow electrolyte to reach active catalytic sites[43,44]. As shown in Supplementary Movies 1–4, bubbles were strongly adsorbed on the surface of electrodes of Mo/C-590 and MoC–$Mo_2C$-690 (Fig. 4a, b), indicating that big bubbles will cover the active sides of catalysts and slow down the mass transfer rate at the interface

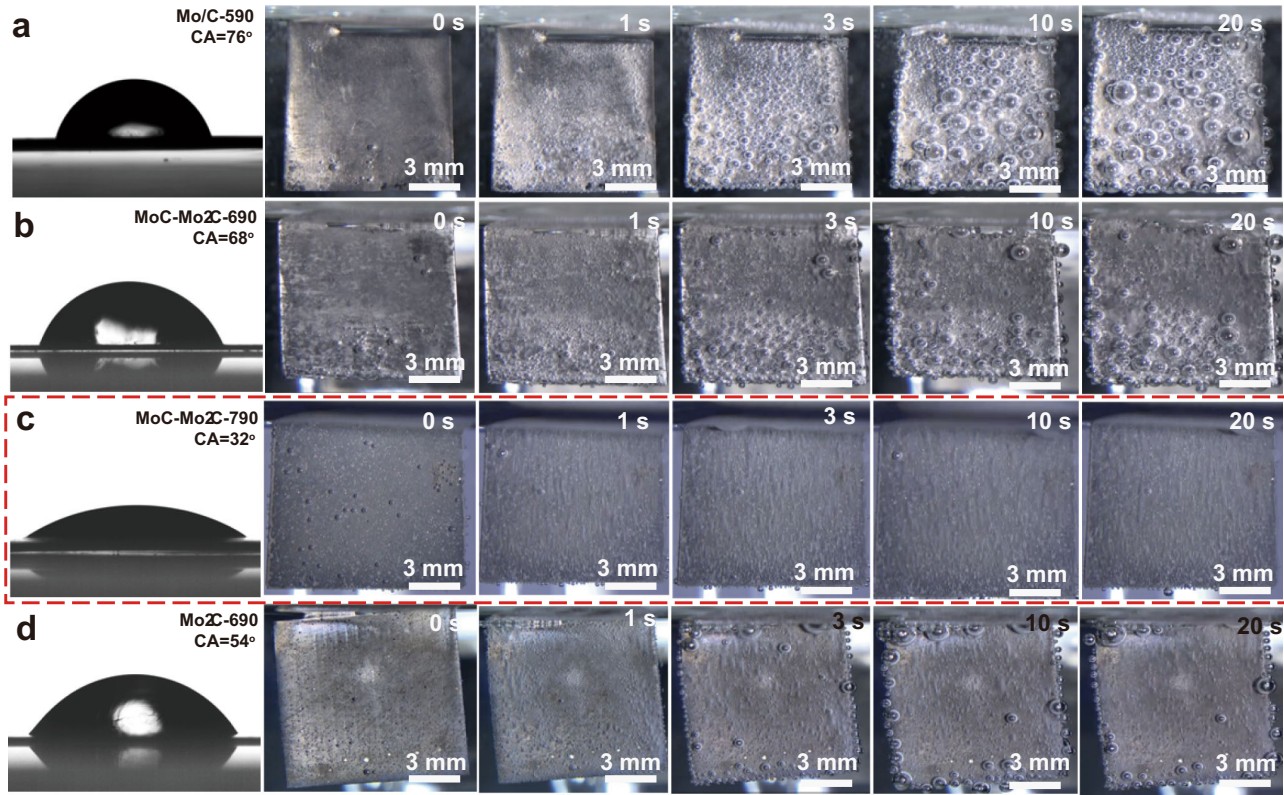

**Fig. 4 Contact angle images between water and electrodes and the bubble evolution process on the surface of electrodes. a** The Mo/C-590 electrode. **b** The MoC–Mo₂C-690 electrode. **c** The MoC–Mo₂C-790 electrode. **d** The Mo₂C-890 electrode.

of electrode/electrolyte. On the electrodes of Mo/C-590 and MoC–Mo₂C-690, small bubbles (<300 μm in diameter) first generate and then grow big large enough (mostly > 800 μm in diameter) and finally leave the electrode smoothly (Supplementary Movies 1–2). At the MoC–Mo₂C-790 and Mo₂C-890 electrodes, H₂ bubbles generate and instantly leave the electrode (Supplementary Movies 3–4). This means that the shielding effect caused by H₂ bubbles is not obvious (Fig. 4c–d) when the size of the bubbles is less than 200 μm in diameter. Constant potential electrolysis (140 mV vs. RHE) further confirms that the MoC–Mo₂C-790 electrode (from 45 mA cm$^{-2}$ to 26 mA cm$^{-2}$) has a slighter shielding effect than that of the MoC–Mo₂C-690 electrode (from 38 mA cm$^{-2}$ to 6 mA cm$^{-2}$) (Supplementary Fig. 34). Thus, the bubbles having a small contact angle on the electrode surface can departure from the electrode more easily, thereby maintaining the active working area and retaining an effective and stable HER activity. For comparison, large bubbles on the Pt electrode (Supplementary Movie 5) may incur the large overpotential of the Pt electrode at high current density (Fig. 2a).

**Scale-up and high-temperature HER performance.** To verify the possibility of scaling up the electrolysis, we prepared a relatively large electrode (3 cm × 11.5 cm) that is 34.5 times that of the previous electrode (Fig. 5a). The contact angle between the as-prepared large-area electrode and water is 36°, which is close to the value between the small MoC–Mo₂C-790 electrode (1 cm × 1 cm) and water. More importantly, the motion behaviors and the size of H₂ bubbles on the large electrode are similar to those that observed on the small electrode (Fig. 5b, Supplementary Movie 6), further confirming the success of scaling up the molten salt electrolysis process. As industrial electrolyzers are usually operated at 60 - 80 °C and 400 to 600 mA cm$^{-2}$, the large

electrode was measured at 70 °C and 500 mA cm$^{-2}$ in both acidic and alkaline solutions (Fig. 5c). The LSV curves before and after stability (Fig. 5d–f) tests keep constant, demonstrating the excellent HER stability even at simulated industrial operating conditions. In addition, a higher operating temperature of 70 °C effectively decreases the HER overpotential of the electrode by ~60 mV and ~70 mV at 500 mA cm$^{-2}$ in acidic and alkaline solutions, respectively (Figs. 2f, 5c, f), which further proves the feasibility of the practical application of the electrolytic MoC-Mo₂C heterojunction electrode in industrial electrolyzers.

**Discussion**

A robust and high-performance self-standing MoC–Mo₂C heterojunction HER electrodes can be prepared on a Mo substrate using CO₂ as the carbon resource and electrons as the reducing agent in molten carbonate through a one-step electrodeposition process. DFT calculations reveal that the superior HER performances of the carbide heterojunction electrode in both acid and alkaline condition are attributed to the MoC (001)–Mo₂C (101) heterojunction that has the $\Delta G_{H^*}$ of −0.13 eV in acidic condition, and the energy barrier 1.15 eV for water dissociation in alkaline solution. In addition, the 3D porous structure is hydrophilic and can suppress the shielding effect of generated H₂ bubbles. Hence, the relatively low overpotentials at high current density (256 mV, 500 mA cm$^{-2}$), good acid- and base-tolerance, ultra-long durability of over 2400 h, good stability at 70 °C, and large-scale electrode demonstration show the possibility of using the self-standing MoC–Mo₂C composite electrode for practical applications. Overall, the use of renewable energy and CO₂ to prepare highly value-added carbon-containing catalysts for various applications provides a sustainable way to reduce the carbon footprint and expedite the utilization of hydrogen energy.

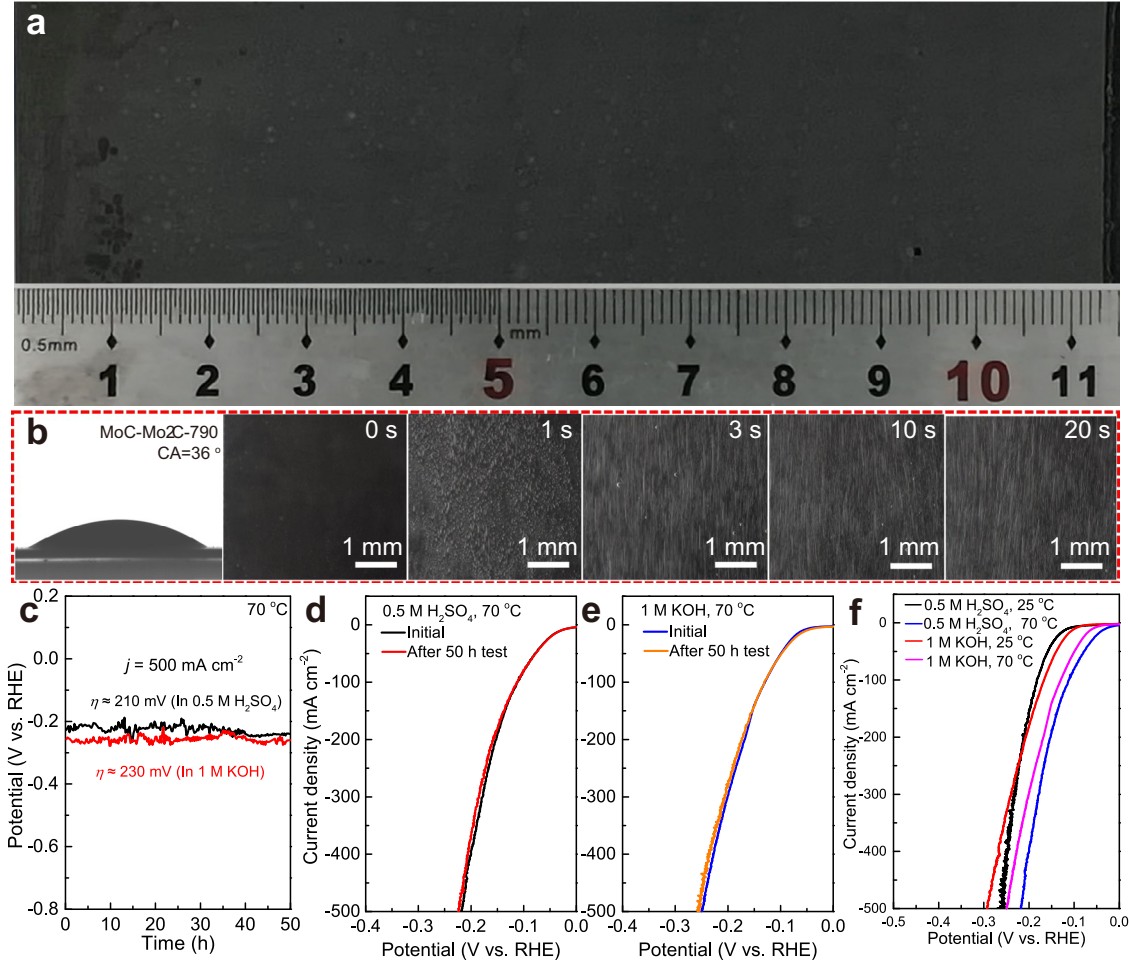

**Fig. 5 Scale-up of the electrolysis and the HER performances of the large electrode operated at high current densities and high operating temperatures. a** Digital image of the large MoC–Mo$_2$C-790 electrode. **b** Contact angle image between water and the large MoC–Mo$_2$C electrodes, and photos showing the H$_2$ bubble evolution process on the electrode. **c** Stability test at simulated industrial operating conditions in both 0.5 M H$_2$SO$_4$ and 1 M KOH at 500 mA cm$^{-2}$ and 70 °C. LSV curves before and after stability tests at simulated industrial operating conditions (**d**, **e**). **f** LSV curves of the MoC-Mo$_2$C-790 electrode in both acidic and alkaline solutions at 25 °C and 70 °C.

## Methods

**Electrodeposition of carbide layer in molten carbonate**. The self-standing MoC–Mo$_2$C heterojunction catalytic layer was electrodeposited on a Mo plate cathode in molten Li$_2$CO$_3$–K$_2$CO$_3$ (molar ratio 1:1) at 590–890 °C. All reagents were of analytical grade (>99%, Sinopharm Chemical Reagent Co., Ltd.). First, Li$_2$CO$_3$ and K$_2$CO$_3$ powders were mixed and placed in an Al$_2$O$_3$ crucible that was contained in a stainless steel (SS) reactor, and the salt mixture was dried at 300 °C for 24 h in a tube furnace. Second, the SS reactor was flowed with CO$_2$ gas and then heated to 590 °C to melt the salt at a ramping rate of 5 °C min$^{-1}$. Third, pre-electrolysis was conducted between a nickel sheet (10 mm × 50 mm) cathode and a home-made Ni–11Fe–10Cu inert anode under a constant cell voltage of 1.5 V for 4 h to remove possible impurities and residual moisture. Fourth, the electrodeposition was performed between a Mo plate (10 mm × 10 mm × 0.1 mm) that was polished with 320 mesh silicon carbide sandpaper and the Ni–11Fe–10Cu inert anode under constant current densities. All electrolysis experiments were carried out at different temperatures at 3.5 mA cm$^{-2}$ for 2 h. After each electrodeposition, the Mo plate cathode was lifted out from the molten salt and cooled down to room temperature in the CO$_2$ atmosphere. Finally, the as-prepared sample was washed by distilled water and ethanol, respectively. The electrodes prepared at different temperatures were named hereafter as Mo/C-590, MoC–Mo$_2$C-690, MoC–Mo$_2$C-790 and Mo$_2$C-890, respectively. More details about the electrodeposition of Mo carbide films and the mechanism of this reaction can be seen in Electronic Supplementary Information (ESI).

**Measurements of HER performances**. HER performances of all electrolytic samples were investigated in a three-electrode cell containing in 0.5 M H$_2$SO$_4$ and 1 M KOH aqueous solution, respectively. The three-electrode cell consisted of a working electrode made from the electrolytic sample, a graphite rod counter

electrode, and a Hg/Hg$_2$SO$_4$ (in H$_2$SO$_4$ solution) or a Hg/HgO (in KOH solution) reference electrode. Linear sweep voltammetry (LSV), cyclic voltammetry (CV) and chronoamperometric tests were performed to evaluate the HER activity and stability of the working electrode on an electrochemical workstation (CH Instrument, CHI1140C, Shanghai Chenhua Instrument Co. Ltd., China). Electrochemical impedance spectroscopy (EIS) measurements were conducted in a frequency range from 100 kHz to 0.01 Hz with an AC potential amplitude of 5 mV on an electrochemical workstation (PARSTA2273, AMETEK Scientific Instruments Co. Ltd., America). Prior to each electrochemical measurement, the electrolyte was bubbled with high-purity Ar for over 40 min to remove the dissolved oxygen in the electrolyte. Unless stated otherwise, all potentials in this paper refer to Reversible Hydrogen Electrode (RHE) by adding a value of (0.656 + 0.059 pH) V in 0.5 M H$_2$SO$_4$ or (0.098 + 0.059 pH) V in 1 M KOH with iR-compensation. The R value was determined by the EIS measurement.

**DFT calculations**. To understand the possible mechanism for hydrogen evolution, we studied the reaction process with DFT calculation. Spin-polarized DFT method was employed for all calculations using the Vienna ab initio Simulation Package (VASP) code[45]. The projector-augmented-wave pseudopotential[46] with a cut-off energy of 400 eV was utilized to treat the core electrons. The Perdew-Burke-Ernzerhof (PBE) version generalized gradient approximation (GGA) functional[47] was used for describing the exchange-correlation energy. The convergence tolerance of energy and force was 10$^{-5}$ eV and 0.01 eV/Å, respectively. The van der Waals interactions were described using the empirical correction in Grimme's scheme (DFT-D3)[48].

The heterojunction interface supercell structure was constructed by stacking Mo$_2$C (101) on top of MoC (001). A 15 Å thick vacuum was inserted to suppress the image interaction induced by a periodic boundary condition. Γ k-point was

adopted for reciprocal space integration. The adsorption energy $E_{ads}$ of H atom on the substrates was defined by[49,50]

$$E_{ads} = E_{H^*} - \left( E_* + \frac{1}{2}E_{H_2} \right) \qquad (1)$$

where $E_{H^*}$, $E_*$, and $E_{H_2}$ denote the total energy of the H atom adsorbed on the substrate, the MoC-Mo$_2$C heterojunction interface, and hydrogen molecule, respectively.

Free energy of the HER intermediate in electrochemical reaction pathways was calculated based on the computational hydrogen electrode (CHE) model proposed by Nørskov et al.[51]. The HER performance was described by the reaction free energy ($\Delta G$), given as follows:

$$\Delta G = \Delta E + ZPE - T\Delta S + \Delta pH \qquad (2)$$

where $\Delta E$ represents the total energy difference between the $E_{H^*}$ and $E_*$, ZPE and $T\Delta S$ are the zero-point energy correction and the entropy change at room temperature (298.15 K). $\Delta pH$ is the free energy correction of $pH$, based on the following equation:

$$\Delta pH = k_B T \times \ln 10 \times pH \qquad (3)$$

**Materials characterizations**. The morphologies and microstructures of all electrodes were characterized by scanning electron microscopy (SEM, MIRA3) equipped with an energy dispersive X-ray spectroscopy (EDS, Aztec Energy), transmission electron microscopy (TEM, FEI tecnai G2 F20), and electron probe micro-analyzer (EPMA-1610, SHIMADZU). Phase compositions of the as-prepared samples were investigated by X-ray diffraction (XRD, Rigaku MiniFlex 600). The valence information of elements in all samples was tested by X-ray photoelectron spectroscopy (XPS ESCALAB 250Xi). The contact angle between electrodes and water was measured on a Dataphysics OCA25 contact angle meter. The interfacial adhesion of the electrolytic film on the surface of the electrode was measured by a WS-2005 scratch tester. The hydrogen evolution process of each electrode was recorded with a SONY FDR-AXP55 camera.

## Data availability

Source data are provided with this paper. Extra data that support the findings of this study are available from the corresponding author upon reasonable request. Source data are provided with this paper.

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

## Acknowledgements

This work was supported by the National Natural Science Foundation of China (5203100, 21673162).

## Author contributions

D.H.W. conceived the idea, H.Y.Y., D.H.W. and W.L. designed the experiment, Y.G., X.T.W. and Z.F.Z. performed the DFT calculations, W.L., F.W. and K.F.D. performed the experiment and analyzed the data, W.L. and H.Y.Y. drafted the manuscript, H.Y.Y., D.H.W., X.T.W., Z.F.G. revised the manuscript and discussed the results, and D.H.W. submitted the paper.

## Competing interests

The authors declare no competing interests.
