## [Peer review file · Nature Communications]

A durable and pH-universal self-standing MoC-Mo₂C heterojunction electrode for efficient hydrogen evolution reactionREVIEWER COMMENTS

Reviewer #1 (Remarks to the Author):

This manuscript reports on the electrochemical deposition of Mo carbide films on Mo substrates from a high temperature melt in which CO₂ has been introduced. The final MoC-MoC₂ material was found to be highly active and stable for the HER over a large pH range which was due to excellent adherence to the Mo substrate, the presence of MoC-MoC₂ heterostructure interfaces and good wettability. Promisingly high current densities could be maintained for up to 100 days which is great to see as often studies limit their lifetime tests to 1 day. Overall this is an excellent piece of work where the material has been characterised thoroughly, the electrochemical performance is outstanding and is backed up by theoretical calculations. I only have some minor comments

Some more detail should be provided on the electrodeposition of Mo carbide films and the mechanism of this reaction. This could be discussed better in the context of previous work where MoC_x films were electrodeposited.

For the XPS – the C 1s spectra should also be shown as well as the O 1s spectra – I assume there is some surface oxidation.

In the experimental a pre-electrolysis step was done with a Ni sheet – why was this done?

It is noted that a NiFeCu anode was used and claimed to be inert – are the authors sure that there is no leaching of these elements? ICP-OES analysis of the catalyst should be undertaken to rule out any trace impurities that may be present in the material.

iR compensation was used – this should be quantified.

Is the electrode tolerant to some oxidation which may occur if integrated directly with an intermittent power supply like solar where reverse current effects can occur?

Reviewer #2 (Remarks to the Author):

In the manuscript, the authors reported a self-standing MoC-Mo₂C catalytic electrode with high catalytic activity and stability in both acidic and alkaline electrolytes. The most impressive data is the long-lasting lifetime of over 2400 h and stability at the industrial operating temperature of ~70 °C. In addition, the outstanding performance was assigned to MoC (001)-Mo₂C (101) heterojunctions and surface hydrophilicity affect H₂ bubbles' motion behaviors. Although there are some other papers published about MoC (001)-Mo₂C (101) heterojunctions as an efficient HER catalyst, I believe this work can be a landmark for non-precious metal-based HER catalytic electrode. Therefore, I would like to suggest publication after a minor revision.

- 1: As the ratio of MoC/Mo₂C changes with the electrolysis temperature, which ratio is most suitable for an efficient HER catalyst?
- 2: The MoC-Mo₂C-790 electrode shows an almost constant catalytic current for 1000 h (2000 h in total) at $\eta = 150$ mV in 0.5 M H₂SO₄ and 1 M KOH, respectively. I still want to confirm that whether it is the same electrode working consecutively in 0.5 M H₂SO₄ and 1 M KOH.
- 3: Please confirm whether the LSV curves in the manuscript are iR- corrected or not?
- 4: For ECSA calculation, why a flat surface is supposed to be ~ 40 $\mu\text{F cm}^{-2}$?
- 5: Why do the MoC/Mo₂C composite electrodes with different electrolysis temperatures have different hydrophilicity?
- 6: As the heterojunctions and surface hydrophilicity both play important roles in HER kinetics, which one is the major factor?

Reviewer #3 (Remarks to the Author):

Active and stable electrocatalysts free-from noble-metal elements are highly desired for hydrogen evolution reactions. Transition-metal carbides showing their promise in this subject have been extensively investigated. This manuscript by Liu et al. reports a self-standing MoC-Mo₂C catalytic electrode prepared via a one-step electro-carbiding approach using CO₂ as the feedstock, which achieves the outstanding HER performances with low overpotentials at 500 mA cm⁻² in both acidic (256 mV) and alkaline electrolytes (292 mV), a long-lasting lifetime of over 2400 h, and high-temperature performance. This outstanding electrocatalytic performance can enable the application in industry. However, the scientific content is not enough for the publication on Nature Communications. The reviewer recommends the rejection. (i) Heterojunction, e.g., MoC-Mo₂C, MoS₂/Mo₂C, Mo₂N/Mo₂C, WO_x/WC, etc., have been previously studied for HER (Chem. Sci. 2016, 7, 3399; Adv. Mater. 2018, 30, 1704156; Nat. Commun. 2019, 10, 269; ACS Energy Lett. 2020, 5, 3560-3568). Moreover, the mechanism interpretation to the enhanced HER herein is ordinary, and cannot bring further insights into the overall kinetics, beyond H* adsorption energy. In particular, alkaline HER also confronts the great barrier of water dissociation that desires comprehensive analysis to identify the influences made by water dissociation energy barriers and H* adsorption energy on the overall reaction rate. In this regard, the novelty of this work is limited. (ii) The feature of this work should be the high current density and long-term durability. Although surface hydrophilicity is highlighted in this work, but the underlying mechanism is still not clear. For example, the relationship between hydrophilicity/aerophobicity and surface porosity, and the merit for mass transport in nanostructured MoC-Mo₂C.

There are some minor issues should be addressed.

- (i) MoC was mentioned with weak *H binding in this work as compared with Mo₂C. This is consistent with previous reports. However, in the DFT calculation (Figures 3b and 3c), MoC shows the much stronger *H binding than that on Mo₂C. Why?
- (ii) Tafel plots were also analyzed in the range with high overpotentials and current density (Page 8 and Figure S6). It's not appropriate because the Tafel analysis is only available in a kinetic-control region. The much higher slope in a range with high overpotentials and current density, in comparison with that in a low-overpotential region, is owing to the logarithmic function. For example, two regions of lg(100) ~ lg(1000) and lg(10) ~ log(100) have the same interval of 1 in Tafel plots, but the current-density range of the former is obviously larger.
- (iii) XPS analysis in Figure S5 is questionable. The Mo²⁺ peaks at 890 eV are quite similar with those of Mo⁰ in the sample at 590 eV. The XPS profiles should be deconvoluted again to make sure the peak positions of a Mo species are consistent in all the samples.

Detailed Responses to the Reviewers

Manuscript ID: NCOMMS-21-10682-T

Title: ‘A self-standing MoC-Mo₂C heterojunction electrode with high electrocatalytic activity, acid and base tolerance, superior hydrophily and stability’

”

Author(s): Wei Liu^{1,2}, Xiting Wang³, Fan Wang^{1,2}, Kaifa Du^{1,2}, Zhaofu Zhang⁴, Yuzheng Guo^{3*}, Huayi Yin^{1,2*} and Dihua Wang^{1,2*}

We are grateful to all the reviewers for their comments and suggestions that are helpful for us to improve our manuscript. We have revised the manuscript taking into full account of the reviewers’ comments and recommendations. Below are our point-to-point responses in the same order as the reviewers’ comments. The changes in the revised manuscript are highlighted in yellow.

REVIEWER COMMENTS

Reviewer #1 (Remarks to the Author):

This manuscript reports on the electrochemical deposition of Mo carbide films on Mo substrates from a high temperature melt in which CO₂ has been introduced. The final MoC-Mo₂C material was found to be highly active and stable for the HER over a large pH range which was due to excellent adherence to the Mo substrate, the presence of MoC-Mo₂C heterostructure interfaces and good wettability. Promisingly high current densities could be maintained for up to 100 days which is great to see as often studies limit their lifetime tests to 1 day. Overall this is an excellent piece of work where the material has been characterised thoroughly, the electrochemical performance is outstanding and is backed up by theoretical calculations. I only have some minor comments.

Response: Thank you very much for your positive recommendations.

1. Some more detail should be provided on the electrodeposition of Mo carbide films and the mechanism of this reaction. This could be discussed better in the context of previous work where Mo_xC films were electrodeposited.

Response: Thanks for the comment. The MoC-Mo₂C electrode was prepared by a one-step electrochemical surface engineering approach in molten $\text{Li}_2\text{CO}_3\text{-K}_2\text{CO}_3$ salts. As shown in Figure 1a of the manuscript, the injected CO_2 was firstly captured by O^{2-} to form soluble CO_3^{2-} ($\text{CO}_2 + \text{O}^{2-} = \text{CO}_3^{2-}$) [Energy Environ. Sci., 2013, 6, 1538-1545]. Then, the CO_3^{2-} got electrons and was reduced to carbon atoms ($\text{CO}_3^{2-} + 4\text{e}^- = \text{C} + 3\text{O}^{2-}$) on the surface of a molybdenum (Mo) plate. At the same time, the released O^{2-} diffused to the inert anode and then discharged to O_2 . Under the synergy of high-temperature molten salt and electric field, the deposited carbon diffused into the Mo plate and simultaneously reacted with Mo atoms spontaneously to generate MoC/Mo₂C ($\text{C} + \text{Mo} = \text{MoC}$, $\Delta G_{T=500-900^\circ\text{C}} = -29.5 \text{ kJ/mol} \sim -30.6 \text{ kJ/mol}$; $\text{C} + 2\text{Mo} = \text{Mo}_2\text{C}$, $\Delta G_{T=500-900^\circ\text{C}} = -47.3 \text{ kJ/mol} \sim -45.7 \text{ kJ/mol}$), forming the MoC-Mo₂C HER electrode finally.

In previous work, most Mo_xC films were prepared using a hydrothermal method by which molybdates were used as the Mo precursor along with a reducing agent to reduce molybdates. The electrolysis was usually conducted in molten halides containing molybdates and carbonates ions as the Mo and C feedstock. However, all electrolyzers used graphite as the anode because it is difficult to employ a low-cost inert anode in molten halides. In previous studies, the aim of making the carbide film is to prepare a protective coating rather than a catalytic layer.

The relative discussion has been added in Electronic Supplementary Information (ESI) on page 2. Thank you again for your suggestion.

2. For the XPS- the C 1s spectra should also be shown as well as the O 1s spectra -I assume there is some surface oxidation.

Response: Thank you very much for the comment. The C1s and O1s XPS spectra of the MoC-Mo₂C-790 electrode surface are shown in Fig. R1 and Supplementary Fig. 1, which indicates that there is some surface oxidation and it is in consistent with the

results in Mo3d XPS spectra (Fig. 1h).

The corresponding XPS analysis of C1s and O1s have been added in Supplementary Fig. 1.

Changes to the revised manuscript. On page 5. “Supplementary Fig. 1a shows the high-resolution spectrum in the C 1s region that was fitted by components corresponding to Mo-C, C-C, C-O, and C=O species. The peak at a BE of 282.8 eV is assigned to the Mo-C species and represents 12 at% of the C1s region. It should be noted that the 12 at% of the C1s region does not mean that the amount of Mo-C is 12 at%, but rather the fact that thin carbon layers formed on the surface of the electrode. The O1s XPS spectrum (Supplementary Fig. 1b) indicates the slight oxidation of the Mo-based composites, which is in consistent with the Mo 3d results in Fig. 1h”.

Fig. R1 (a) C1s XPS spectra and (b) O1s XPS spectra of the MoC-Mo₂C-790 electrode surface.

3. In the experimental a pre-electrolysis step was done with a Ni sheet – why was this done?

Response: Thank you very much for your question. A pre-electrolysis experiment is a common method to remove impurities and residual moisture of the molten salt electrolyte. The corresponding revision has been made in the part of “Electrodeposition of carbide layer in molten carbonate”.

Changes to the revised manuscript. On page 19. “Third, pre-electrolysis was

conducted between a nickel sheet (10 mm × 50 mm) cathode and a home-made Ni-11Fe-10Cu inert anode under a constant cell voltage of 1.5 V for 4 h to remove possible impurities and residual moisture”.

4. It is noted that a NiFeCu anode was used and claimed to be inert – are the authors sure that there is no leaching of these elements? ICP-OES analysis of the catalyst should be undertaken to rule out any trace impurities that may be present in the material.

Response: Thank you very much for your valuable questions and suggestions. According to our previous work about the Ni11Fe10Cu anode [Electrochimica Acta, 2018, 279:250-257; Corrosion Science, 141, 2018, 168–174; Corrosion Science, 2016, 112:54-62], the released O²⁻ from the reduction of CO₃²⁻ diffused to the inert anode and discharged to O₂. And the formed O₂ reacted with Ni11Fe10Cu to form a protective oxide scale, which is able to prevent the Ni11Fe10Cu substrate from leaching of some elements for at least 600 h. The preparation of the Mo_xC film in this paper only took 2 hours. The inert anode was quite stable according to the weight and dimension change of the anode. In order to further confirm the stability of the anode, we analyzed the impurities in the Mo_xC film.

We conducted the ICP-OES test of the Mo_xC electrode, and the results are shown in the following Table R1. The concentrations of Cu, Fe and Ni are all below 6 mg/kg, thus the effect on the HER performance is almost ignorable.

Table R1. The content of Mo, Ni, Fe and Cu in the MoC-Mo₂C-790 electrode measured by ICP-OES.

Elements	Sample concentration (mg/kg)
Cu	4.0192
Fe	3.5116
Ni	5.4241
Mo	980321.6369

5. iR compensation was used – this should be quantified.

Response: Thank you very much for your suggestion. The R value used for the iR compensation was determined by EIS measurement. We have added this point in the Method part of the revised manuscript as well as in the figure captions.

The corresponding LSV curves of different electrodes before and after iR correction in 0.5 M H₂SO₄ and 1 M KOH are presented in the following figure (Fig. R2). For the convenience of readers, we have added the figure in the Electronic Supplementary Information (ESI) (Supplementary Fig. 7).

Fig. R2 (a) LSV curves of different electrodes before and after iR correction in 0.5 M H₂SO₄. (b) LSV curves of different electrodes before and after iR correction in 1.0 M KOH.

6. Is the electrode tolerant to some oxidation which may occur if integrated directly with an intermittent power supply like solar where reverse current effects can occur?

Response: Many thanks for the interesting question. Based on the comment, we tested the open circuit potentials (corrosion potential) of the MoC-Mo₂C-790 electrode and its anodic polarization curves in 0.5 M H₂SO₄ and 1.0 M KOH, respectively. As shown in Fig. R3, the electrode had a stable potential profile in both 0.5 M H₂SO₄ and 1.0 M KOH, which indicates that the MoC-Mo₂C coating has a good oxidation resistance at the open circuit. While the anodic polarization curves show an oxidation current upon anodic polarization starting from the open circuit potential. Thus, the electrode cannot withstand a strong anodic polarization for a long time. The stability of the Mo_xC electrode under anodic potential depends on its intrinsic physicochemical properties. For the practical application, it should be avoided to reverse the

polarization direction.

Fig. R3 (a) Open circuit potential profiles as a function of time of the MoC-Mo₂C-790 electrode in 0.5 M H₂SO₄ and 1.0 M KOH. (b) Anodic polarization curves of the MoC-Mo₂C-790 electrode in 0.5 M H₂SO₄ and 1.0 M KOH.

Reviewer #2 (Remarks to the Author):

In the manuscript, the authors reported a self-standing MoC-Mo₂C catalytic electrode with high catalytic activity and stability in both acidic and alkaline electrolytes. The most impressive data is the long-lasting lifetime of over 2400 h and stability at the industrial operating temperature of ~70 °C. In addition, the outstanding performance was assigned to MoC (001)-Mo₂C (101) heterojunctions and surface hydrophilicity affect H₂ bubbles' motion behaviors. Although there are some other papers published about MoC (001)-Mo₂C (101) heterojunctions as an efficient HER catalyst, I believe this work can be a landmark for non-precious metal-based HER catalytic electrode. Therefore, I would like to suggest publication after a minor revision.

Response: Thank you very much for your positive recommendations.

1. As the ratio of MoC/Mo₂C changes with the electrolysis temperature, which ratio is most suitable for an efficient HER catalyst?

Response: Thanks for the good question. As shown in Fig. 1g, Fig. 2a and Fig. 2c, the MoC -Mo₂C-790 HER electrode (with 65.4% of MoC and 34.6% of Mo₂C) with an

optimal $\text{Mo}^{3+}/\text{Mo}^{2+}$ ratio (Supplementary Table 6, Supplementary Fig. 20) has the highest catalytic activity among the four measured electrodes. There might exist a more optimistic ratio of $\text{MoC}/\text{Mo}_2\text{C}$ and it remains to be investigated in future.

2. The $\text{MoC}-\text{Mo}_2\text{C}-790$ electrode shows an almost constant catalytic current for 1000 h (2000 h in total) at $\eta = 150$ mV in 0.5 M H_2SO_4 and 1 M KOH, respectively. I still want to confirm that whether it is the same electrode working consecutively in 0.5 M H_2SO_4 and 1 M KOH.

Response: Thanks for the question and sorry for not making this point clear in the original manuscript. It is the same electrode working consecutively in 0.5 M H_2SO_4 and 1 M KOH.

3. Please confirm whether the LSV curves in the manuscript are iR- corrected or not?

Response: The LSV curves in the manuscript had been iR-corrected. The corresponding LSV curves of different electrodes before and after iR correction in 0.5 M H_2SO_4 and 1 M KOH is presented below and Supplementary Fig. S7 in the revised manuscript. The value of R was determined by the EIS measurement for each experiment, we have added this point in the Method part.

Fig. R4 (a) LSV curves of different electrodes before and after iR correction in 0.5 M H_2SO_4 . (b) LSV curves of different electrodes before and after iR correction in 1.0 M KOH.

4. For ECSA calculation, why a flat surface is supposed to be $\sim 40 \mu\text{F cm}^{-2}$?

Response: Typically, the non-Faradic capacitance for the electrode/solution interface in aqueous solution is in the range of 20-60 $\mu\text{F cm}^{-2}$ (Electrochim. Acta, 2002, 47, 3571-3594; ACS Catal., 2012, 2, 1916-1923; Angew.Chem. Int. Ed., 2014, 53, 14433-14437; Energy Environ. Sci., 2016, 9, 1468-1475), 40 $\mu\text{F cm}^{-2}$ was usually used to calculate the ECSA and turnover frequency (TOF).

5. Why do the MoC/Mo₂C composite electrodes with different electrolysis temperatures have different hydrophilicity?

Response: Thanks for the question. Fig. 4 indicates that the MoC/Mo₂C composite electrodes prepared at different electrolysis temperatures have different hydrophilicity. The main reason is due to the difference of surface morphology between the electrodes prepared at different temperatures. From Fig. 1b and Supplementary Fig. 3, it can be known that the electrodes show different surface morphologies with different roughness and porous characteristics. When a liquid comes in contact with a textured surface, it leads to either the fully wetted Wenzel (Ind. Eng. Chem. 1936, 28, 988) state or the Cassie-Baxter (Trans. Faraday Soc. 1944, 40, 0546) state, which supports a composite solid-liquid-air interface. For the hydrophilic electrodes (Contact angle < 90 °), the calculation of the contact angle (θ_r) between electrodes and water should employ the following equation:

$$\cos \theta_r = r \cos \theta$$

where θ_r represents the actual measured contact angle. r represents the roughness factor, which is obtained by dividing the actual area by the geometric area. θ is the contact angle of an ideal smooth surface.

Therefore, when the contact angle of an ideal smooth surface is < 90 °, a larger r value corresponds to a smaller θ_r . According to the ECSA of different electrodes in Table S7, the r value of MoC-Mo₂C -790 > the r value of Mo₂C -890 > the r value of MoC-Mo₂C -690 > the r value of M/C-590. So, the θ_r value of MoC-Mo₂C -790 < the θ_r value of Mo₂C -890 < the θ_r value of MoC-Mo₂C -690 < the θ_r value of M/C-590.

In fact, there are many similar conclusions that a rough structures with roughness at both the micro- and nano-scale not only generate a strong capillary force to pump

liquid, but also reduce interfacial adhesion to facilitate gas bubble release (Adv. Mater. 2012, 24, 5838–5843; Adv. Energy Mater. 2018, 8, 1802445; Angew. Chem. Int. Ed. 2015, 54, 4876–4879; Soft Matter 2012, 8(7): 2261).

6. As the heterojunctions and surface hydrophilicity both play important roles in HER kinetics, which one is the major factor?

Response: Thanks for the insightful question. Yes, both of the heterojunctions and surface hydrophilicity play important roles in HER kinetics, but their contribution is different under the different working current densities. The MoC-Mo₂C heterojunctions play important roles in controlling the thermodynamics of the HER, and the surface hydrophilicity of electrodes relates to the kinetics of the HER at a high current density. It can be seen from the LSV curve of the Pt electrode in Fig. 2a that the Pt electrode outperforms all HER electrodes including the MoC-Mo₂C-790 electrode at 0-200 mA cm⁻². But the performance of the Pt electrode is inferior to the MoC-Mo₂C-790 electrode when the current density exceeds 200 mA cm⁻². In addition, the LSV curve of the Pt electrode shows obvious fluctuations when the current density exceeds 200 mA cm⁻² because of the shielding effect caused by the generated bubbles at the surface of Pt electrode. The shielding effect has been directly observed from the Video S5. In the thermodynamic point of view, Pt has an optimal ΔG_{H^*} , so it has the smallest hydrogen evolution overpotential.

Fig. R5 Polarization curves (with iR correction) of various electrodes in 0.5 M H₂SO₄.

Reviewer #3 (Remarks to the Author):

Active and stable electrocatalysts free-from noble-metal elements are highly desired for hydrogen evolution reactions. Transition-metal carbides showing their promise in this subject have been extensively investigated. This manuscript by Liu et al. reports a self-standing MoC-Mo₂C catalytic electrode prepared via a one-step electro-carbiding approach using CO₂ as the feedstock, which achieves the outstanding HER performances with low overpotentials at 500 mA cm⁻² in both acidic (256 mV) and alkaline electrolytes (292 mV), a long-lasting lifetime of over 2400 h, and high-temperature performance. This outstanding electrocatalytic performance can enable the application in industry. However, the scientific content is not enough for the publication on Nature Communications. The reviewer recommends the rejection. (i) Heterojunction, e.g., MoC-Mo₂C, MoS₂/Mo₂C, Mo₂N/Mo₂C, WO_x/WC, etc., have been previously studied for HER (Chem. Sci. 2016, 7, 3399; Adv. Mater. 2018, 30, 1704156; Nat. Commun. 2019, 10, 269; ACS Energ. Lett. 2020, 5, 3560-3568). Moreover, the mechanism interpretation to the enhanced HER herein is ordinary, and cannot bring further insights into the overall kinetics, beyond H* adsorption energy. In particular, alkaline HER also confronts the great barrier of water dissociation that desires comprehensive analysis to identify the influences made by water dissociation energy barriers and H* adsorption energy on the overall reaction rate. In this regard, the novelty of this work is limited. (ii) The feature of this work should be the high current density and long-term durability. Although surface hydrophilicity is highlighted in this work, but the underlying mechanism is still not clear. For example, the relationship between hydrophilicity/aerophobicity and surface porosity, and the merit for mass transport in nanostructured MoC-Mo₂C.

Response: Many thanks to the reviewer for your comments on the merit of our work that “This outstanding electrocatalytic performance can enable the application in industry”. We are sorry that the reviewer proposed a high scientific standard here by comparing our work with four kinds of catalysts (MoC-Mo₂C, MoS₂/Mo₂C, Mo₂N/Mo₂C, WO_x/WC, etc.) which we cited in our original manuscript (Chem. Sci.

2016, 7, 3399; Adv. Mater. 2018, 30, 1704156; Nat. Commun. 2019, 10, 269; ACS Energ. Lett. 2020, 5, 3560-3568). In the process of preparing this paper, we had carefully dug into these literatures and believed our work is a significant advancement in this field.

Among tons of papers regarding carbide-based HER electrodes, we selected these papers because these papers inspired us a lot and we think that our work makes important progresses on developing a straightforward synthetic method, achieving a long service time, revealing the thermodynamic mechanism using DFT calculations, and uncovering the surface properties that controlled the kinetics of the HER under high current densities.

We are sorry that we have not compared our work with the above-mentioned papers in more detail, which leads to an impression that our work is lack of scientific advances compared with previous works. We also thank the editor for giving us a chance to revise our manuscript. We really appreciate this opportunity to explain both scientific and technical advances of our work and the valuable time the reviewer reexamined our work.

First, we read the four papers carefully again, summarized the keys points of these work, and pointed out the different points with our work.

MoC-Mo₂C: This paper reported the MoC-Mo₂C heteronanowires prepared by the carbonization of various MoO_x-amine precursors. The HER activity of this powdery heteronanowires catalyst was measured to be $\eta=126$ at a current density of 10 mA cm⁻². This paper reported its HER activity at a low current density (10 mA cm⁻²). However, the underlying mechanism was not studied in detail. In addition, the stability of the heteronanowire catalyst was tested for only 40 h, and it had a low current retention rate of about 50% in KOH after 20 h.

Thus, the preparation method, service time, and working conditions are quite different from our work. It is not a self-standing electrode. In addition, theoretical calculation and surface properties were not investigated in this work.

MoS₂/Mo₂C: First, the composition of the catalyst is different from ours. This paper introduced the MoS₂/Mo₂C heterojunction, which exhibited excellent HER activity at a high current density ($\eta=225$ V at 1000 mA cm⁻²). However, this electrode was tested for the stability for a total of 48 hours, which is much shorter than our work. And the mechanism of electrode stability was not investigated. In addition, the preparation process contained two steps .

Thus, the composition of the catalysts, service time, working conditions, synthetic method, Mo precursor are different from our work.

Mo₂N/Mo₂C: This paper reported the Mo₂N-Mo₂C heterojunction. The author highlighted that the N-Mo-C interface with the addition of graphene oxide had the smallest ΔG_{H^*} value (0.046 eV). However, the measured HER activity of the Mo₂N-Mo₂C heterojunction was relatively low ($\eta=300$ mV at 200 mA cm⁻² in 0.5 M H₂SO₄, $\eta=530$ mV at 200 mA cm⁻² in 1 M KOH). And the mechanism of electrode stability was not investigated.

Thus, the composition of the catalysts, service time, working conditions, synthetic method, Mo precursor are different from our work. In addition, surface properties were not investigated in this work.

WO_x/WC: The paper reported the WO_x/WC surface heterojunction catalyst, and demonstrated that the WO_x/WC surface heterojunction catalyst had a more excellent HER activity than that of pure WO_x and WC. DFT calculations and in situ XAS were employed to reveal the mechanistic process. Even though the WO_x/WC surface heterojunction had a perfect Gibbs free energy of H* adsorption ($\Delta G_{H^*}=0.09$), the corresponding HER activity was not so high ($\eta=233$ mV at 20 mA cm⁻² in 0.5 M H₂SO₄). The paper highlighted the influence of WO_x/WC surface heterojunction on the intrinsic catalytic activity of the material, but the performance at a high current density was not performed, and the surface properties were not studied.

Thus, the composition of the catalysts, service time, working conditions, and

synthetic method are different from our work. In addition, the WO_x/WC is not a self-standing electrode and surface properties are not investigated in this work.

In summary, only one paper reported the MoC-Mo₂C heterojunction and the result is preliminary. They prepared the MoC-Mo₂C powder that was then coated on a glass carbon electrode, which was not a self-standing electrode. Our group has worked in the field of molten salt electrochemistry for more than 20 years. The electrified synthesis and efficient conversion and utilization of CO₂ have gained increasing attention in recent years. After many years of hard work, we try to combine the molten salt electrolysis with the surface engineering technology to make functional layers for green energy conversions. This work was done based on previous establishments such as inert anode development, CO₂ capture and reduction, physicochemical properties of molten salts, and fundamentals of electrochemistry and materials. In addition, we try our best to scale up this process that a larger electrode (3 cm×11.5 cm) electrode was successfully prepared. And a working condition similar to the industrially deployed electrolyzer was applied to measure the HER performance. To reveal the underlying mechanism, we worked with other groups at Wuhan University and Cambridge University to do the theoretical calculations. In this regard, we hope our work can make contributions to both scientific advances and technical progresses.

Second, we double-checked the DFT calculation. Our work indicates that both surface chemical composition and surface microstructure of catalysts should be considered toward HER activity and stability at high current densities and high temperatures, which is a general requirement for developing high-performance HER catalysts. Regarding the surface chemistry, our results indicate that the 64.5%MoC-34.5%Mo₂C heterojunction improves the Gibbs free energy of H* adsorption ($\Delta G_{\text{H}^*} = -0.13$) and the wettability between the electrode and water of pure MoC or Mo₂C by changing the electronic structures. Regarding the surface microstructure, the abundant 3D honeycomb-like structures not only provide a super large specific surface area, but

also provide an effective channel for the rapid entry of water and the rapid removal of gas, which is conducive to the improvement of electrode surface hydrophilicity and reduce the erosion effect of gas on the surface-active material of the electrode, improving the HER activity and stability of catalysts. Both of this kind of structures and the mechanism we found is instructive for the design of other catalytic electrodes, *e.g.*, metal carbides, metal nitrides, and MXenes. Besides, the preparation of HER electrodes by molten salts electrolysis is reported for the first time, which is simple, green and effective, and also provides an instructive method for the preparation of other HER catalysts, such as metal carbides, metal nitrides, metal sulfide, etc. Therefore, we believe our manuscript offers new insights into the HER mechanism and provides a general guideline for catalyst design.

Third, we agree that the relationship between hydrophilicity/aerophobicity and surface porosity can add scientific value of this work. Fig. 4 indicates that the MoC/Mo₂C composite electrodes prepared at different electrolysis temperatures have different hydrophilicity. In fact, this could be caused by the different surface morphologies of electrodes prepared at different temperatures. From Fig. 1b and Supplementary Fig. 3, it can be known that different HER electrodes show different surface morphologies with different porous characteristics and roughness. Thus, the HER electrodes prepared at different temperatures exhibit different surface roughness inevitably.

When a liquid comes in contact with a textured surface, it leads to either the fully wetted Wenzel (Ind. Eng. Chem. 1936, 28, 988) state or the Cassie-Baxter (Trans. Faraday Soc. 1944, 40, 0546) state, which supports a composite solid-liquid-air interface. For the hydrophilic electrodes (Contact angle < 90 °), the calculation of the contact angle (θ_r) between electrodes and water should employ the following equation:

$$\cos \theta_r = r \cos \theta$$

where θ_r represents the actual measured contact angle. r represents the roughness factor, which is obtained by dividing the actual area by the geometric area. θ is the

contact angle of ideal smooth surface.

Therefore, when the contact angle of an ideal smooth surface is $< 90^\circ$, the larger the r value, the smaller the θ_r . According to the ECSA of different electrodes in Table S7, the r value of MoC-Mo₂C -790 $>$ the r value of Mo₂C -890 $>$ the r value of MoC-Mo₂C -690 $>$ the r value of M/C-590. So, the θ_r value of MoC-Mo₂C -790 $<$ the θ_r value of Mo₂C -890 $<$ the θ_r value of MoC-Mo₂C -690 $<$ the θ_r value of M/C-590.

In fact, there are many similar conclusions that structures with roughness at both the micro- and nano-scale not only generate a strong capillary force to pump liquid, but also reduce interfacial adhesion to facilitate gas bubble release (Adv. Mater. 2012, 24, 5838–5843; Adv. Energy Mater. 2018, 8, 1802445; Angew. Chem. Int. Ed. 2015, 54, 4876–4879; Soft Matter 2012, 8(7): 2261).

Until now, we can only provide this information about the surface properties of the as-prepared electrode. This is a good question need to be uncovered in our future work with more advanced characterization and theoretical methods.

There are some minor issues should be addressed.

1. MoC was mentioned with weak H* binding in this work as compared with Mo₂C. This is consistent with previous reports. However, in the DFT calculation (Figures 3b and 3c), MoC shows the much stronger H* binding than that on Mo₂C. Why?

Response: We thank the reviewer for helping us to clear this point. According the DFT calculation, MoC has the much stronger H* binding than that on Mo₂C indeed. We may not express it clearly and we have made corresponding revisions in manuscript. The strong Mo-H on Mo₂C benefits H⁺ reduction (*i.e.*, the Volmer step), but limits the desorption process of adsorbed H (H_{ads}) (*i.e.*, the Heyrovsky step). And the MoC-Mo₂C-790 electrode with the formed MoC-Mo₂C heterojunctions weaken the strength of Mo-H towards accelerating H_{ads} desorption, and thus significantly enhances the HER performance.

Changes to the revised manuscript. On Page 12. “The reduction of electron density

in the MoC-Mo₂C-790 electrode for the formation of MoC-Mo₂C heterojunction with the ascendency of Mo³⁺ weakens the strength of Mo-H towards accelerating H_{ads} desorption, and thus significantly enhances the HER performance. With increasing n_{3+/2+} to 1.98, the MoC-Mo₂C-690 electrode with a main MoC phase shows an obviously decreased HER activity comparing with the MoC-Mo₂C-790 electrode”.

2. Tafel plots were also analyzed in the range with high overpotentials and current density (Page 8 and Figure S6). It's not appropriate because the Tafel analysis is only available in a kinetic-control region. The much higher slope in a range with high overpotentials and current density, in comparison with that in a low-overpotential region, is owing to the logarithmic function. For example, two regions of lg(100) ~ lg(1000) and lg(10) ~ log (100) have the same interval of 1 in Tafel plots, but the current-density range of the former is obviously larger.

Response: Thanks for the comment. This is a goof point. Yes, we agree that the Tafel analysis is only available in a kinetic-control region. So we have changed the expression by using “slope”, instead of “Tafel slope” in Supplementary Fig. 8. The purpose of Supplementary Fig. 8 is to show how much potential is needed when increasing the current to an industry-scale current, which could be an indicator to evaluate the performance of a catalyst working at large current densities. In fact, the literature we mentioned (Nat.Comm. 2019, 10, 269) also used the similar calculation to evaluate the performance of a catalyst at large current densities. Of course, if the reviewer still thinks it is not proper, we can further revise or delete Supplementary Fig. 8 based on your suggestion.

Changes to the revised manuscript. On Page 8. “Although the Tafel slope value of the MoC-Mo₂C-790 electrode is slightly higher than that of Pt (35 mV dec⁻¹), the slope value of the MoC-Mo₂C-790 (110 mV dec⁻¹) electrode at the strong polarization zone (at 200 to 500 mA cm⁻²) is much smaller than that of Pt (405 mV dec⁻¹) (Supplementary Fig. 8). This means that the MoC-Mo₂C-790 electrode is able to operate at high current densities with a smaller overpotential, which could be an indicator to evaluate the performance of a catalyst at large current densities and is

meaningful for practical use”.

3. XPS analysis in Figure S5 is questionable. The Mo^{2+} peaks at 890 °C are quite similar with those of Mo^0 in the sample at 590 °C. The XPS profiles should be deconvoluted again to make sure the peak positions of a Mo species are consistent in all the samples.

Response: Thank the reviewer very much for the comment. We have carried out the XPS test of the Mo_2C -890 electrode again, and the corresponding spectrum can be seen in Fig. R6 and Supplementary Fig. 6.

Fig. R6 XPS patterns of different electrolytic electrodes prepared at 590 °C, 690 °C and 890 °C.

REVIEWER COMMENTS

Reviewer #1 (Remarks to the Author):

The authors have thoroughly addressed all the reviewer comments

Reviewer #2 (Remarks to the Author):

As all my concerns and questions have been well addressed by the authors, I am very satisfied with the explanation and revisions made accordingly. However, after going over the other referee's comments, I suggest the authors amend DFT calculation of the water dissociation step which can better explain the mechanism of HER activity in alkaline electrolytes.

Reviewer #3 (Remarks to the Author):

The authors have thoroughly revised the manuscript and provided clear response in details that addressed my concerns. I think this manuscript is acceptable for the publication on Nat. Commun.

Detailed Responses to the Reviewers

Manuscript ID: NCOMMS-21-10682B

Title: A self-standing MoC-Mo₂C heterojunction electrode with high electrocatalytic activity, acid and base tolerance, superior hydrophily and stability

Author(s): Wei Liu^{1,2}, Xiting Wang³, Fan Wang^{1,2}, Kaifa Du^{1,2}, Zhaofu Zhang⁴, Yuzheng Guo^{3*}, Huayi Yin^{1,2*} and Dihua Wang^{1,2*}

We are grateful to all the reviewers and editors for their comments and suggestions that are helpful for us to improve our manuscript. We have revised the manuscript taking into full account of the reviewers' and editor's comments and recommendations. Below are our point-to-point responses in the same order as the reviewers' comments. The changes in the revised manuscript are highlighted in yellow.

REVIEWER COMMENTS

Reviewer #1 (Remarks to the Author):

The authors have thoroughly addressed all the reviewer comments

Response: Thank you very much for your positive recommendations. And we are also grateful for your comments and suggestions that are helpful for us to improve our manuscript.

Reviewer #2 (Remarks to the Author):

As all my concerns and questions have been well addressed by the authors, I am very satisfied with the explanation and revisions made accordingly. However, after going over the other referee's comments, I suggest the authors amend DFT calculation of the water dissociation step which can better explain the mechanism of HER activity in alkaline electrolytes.

Response: Thank you very much for your positive recommendations and this kind

suggestion. According to your suggestion, we performed the following calculations and the results are shown in Fig. R1 and Supplementary Fig. 24.

The water dissociation step, considered as the key rate determining step, is calculated to further explain HER mechanism in alkaline condition, where water supplies hydrogen [Energy Environ. Sci. 7, 2255-2260 (2014), Science 334, 1256-1260 (2011)]. To be specific, the H₂O molecule would adsorb one electron to be dissociated into intermediate *H and *OH as shown in Fig. R1 and Supplementary Fig. 24. The water molecule would adsorb on the Pt atom and break into intermediate *H and *OH with an energy barrier of 0.97 eV, consistent with previous work of Pt (111) surface in alkaline condition [Nature Comm. 8: 14580 (2017)]. Similarly, the dissociation of water on the Mo₂C-MoC interface also happens on Mo site and shares the same reaction path. It can be known by calculation that the Mo₂C-MoC interface possesses the water dissociation barrier of 0.32 eV, which is much lower than that of Pt (111) (0.97 eV). The Mo₂C-MoC interface can promote the dissociation of water and the energy barrier is comparable to the previously reported PtNi alloy system [Nature comm. 8: 14580 (2017)].

Fig. R1 Calculated ΔG_{H^*} diagram of the HER at the equilibrium potential in alkaline electrolyte.

Reviewer #3 (Remarks to the Author):

The authors have thoroughly revised the manuscript and provided clear response in details that addressed my concerns. I think this manuscript is acceptable for the publication on Nat. Commun.

Response: Thank you very much for your positive recommendations. And we are also grateful for your comments and suggestions that are helpful for us to improve our manuscript.

REVIEWER COMMENTS

Reviewer #2 (Remarks to the Author):

As the amended DFT calculation can well explain the water dissociation step, I have no further questions or comments. Therefore, I am very happy to suggest acceptance for publication.

Reviewer #4 (Remarks to the Author):

The author reported a MoC-Mo₂C heterojunction structure exhibiting high HER activity and explored DFT calculation to give explanation. However, in my opinion, much more work need to be done for the DFT section.

Firstly, due to the reaction occurring on alkaline condition, it is very necessary to consider the energy barriers of two steps of water dissociation. The authors only compared the first water dissociation energy barrier of Mo₂C-MoC with Pt and dropped this section in SI. What the authors need to do is to compare the water dissociation energy barrier on Mo₂C, MoC, and Mo₂C-MoC, respectively. And what's more, not only the first water dissociation step, but the second water dissociation step is also supposed to be done which is usually the rate-determining step.

Secondly, the authors still took the ΔG_H^* value as main descriptor of HER activity in Figure 3. But for alkaline HER, the smaller ΔG_H^* value usually corresponds with the higher water dissociation energy barrier. Then, the high alkaline HER performance is supposed to correspond with the moderate ΔG_H^* value, not close to zero. In addition, the author mentioned "As illustrated in Fig.3g, the MoC-Mo₂C heterojunction exhibits better HER kinetics compared to those of bare MoC and bare Mo₂C. " While, I cannot get any HER kinetics information from Fig. 3g. So, Fig.3 needs to be reorganized where some figures are not valuable anymore.

Finally, the calculated results by the software are just like the experimental measurement by the instrument. The authors need to give further theoretical analysis why the heterojunction structure exhibits high HER kinetics. Since structure determining the performance, what kind of intrinsic electronic structure corresponds with the heterojunction geometric structure and thus results in the high HER activity of Mo₂C-MoC, not Mo₂C or MoC. Having understanding this benefits the significance of this article and will be very helpful for readers to find more high-performance HER catalysts.

Detailed Responses to the Reviewers

Manuscript ID: NCOMMS-21-10682B

Title: ‘A self-standing MoC-Mo₂C heterojunction electrode with high electrocatalytic activity, acid and base tolerance, superior hydrophily and stability’

”

Author(s): Wei Liu^{1,2}, Xiting Wang³, Fan Wang^{1,2}, Kaifa Du^{1,2}, Zhaofu Zhang⁴, Yuzheng Guo^{3*}, Huayi Yin^{1,2*} and Dihua Wang^{1,2*}

We are grateful to the reviewers for their comments and suggestions that are helpful for us to improve our manuscript. We have revised the manuscript taking into full account of the reviewers’ comments and recommendations. Below are our point-to-point responses in the same order as the reviewers’ comments. The changes in the revised manuscript are highlighted in yellow.

Reviewer #2 (Remarks to the Author):

As the amended DFT calculation can well explain the water dissociation step, I have no further questions or comments. Therefore, I am very happy to suggest acceptance for publication.

Response: We thank the reviewer very much for the positive recommendation.

Reviewer #4 (Remarks to the Author):

The author reported a MoC-Mo₂C heterojunction structure exhibiting high HER activity and explored DFT calculation to give explanation. However, in my opinion, much more work need to be done for the DFT section.

Response: Many thanks for your overall positive comments. As suggested, we have done more DFT calculations to explain the high HER activity of the electrode in alkaline solution and have added new results of DFT calculation in the revised manuscript.

1. Firstly, due to the reaction occurring on alkaline condition, it is very necessary to consider the energy barriers of two steps of water dissociation. The authors only compared the first water dissociation energy barrier of Mo₂C-MoC with Pt and dropped this section in SI. What the authors need to do is to compare the water dissociation energy barrier on Mo₂C, MoC, and Mo₂C-MoC, respectively. And what's more, not only the first water dissociation step, but the second water dissociation step is also supposed to be done which is usually the rate-determining step.

Response: In this paper, we reported the super HER catalytic activity of the self-standing MoC-Mo₂C electrode under both acid and alkaline conditions and try to explain its performances assisted by theoretical calculations. We agree with the reviewer that the comparison of energy barriers of water dissociation on Mo₂C, MoC, and MoC-Mo₂C and the two steps of water dissociation is helpful to explain the results. Following the suggestion, we calculated and compared the water dissociation energy barriers on Mo₂C, MoC, and Mo₂C-MoC. The results are shown in Fig. 3c in the revised manuscript and the following Fig. R1 (The reaction energy diagram of water dissociation on MoC, Mo₂C, and MoC-Mo₂C). It can be seen that the energy barrier of

the second H₂O dissociation step on the MoC-Mo₂C heterojunction is 1.15 eV, while it is 1.90 and 3.33 eV for bare MoC and Mo₂C, respectively. These results indicate that the MoC-Mo₂C has a better intrinsic alkaline HER activity than that of pure MoC and Mo₂C, which is consistent with our experimental results. Many thanks for the kind suggestion.

In the revised manuscript, we have added more discussion for the alkaline condition. Page 13, line 261: “During the process of HER in alkaline solution, the first H₂O dissociates into intermediate H* and OH*. Then, the dissociation of the second H₂O leads to the generation of H₂. The HER energy diagrams were plotted to explain the HER performance in alkaline solution. As shown in Fig. 3c, the highest energy barrier (the second H₂O dissociation step) of the MoC-Mo₂C heterojunction is 1.15 eV, while the HER performance on bare MoC and Mo₂C is relatively poor, with the highest energy barrier of 1.90 eV and 3.33 eV on bare MoC and Mo₂C, respectively.”

Fig. R1 Reaction energy diagram of water dissociation on MoC, Mo₂C, and MoC-Mo₂C, including the two steps of water dissociation, in alkaline solution.

2. Secondly, the authors still took the ΔG_{H^*} value as main descriptor of HER activity in Figure 3. But for alkaline HER, the smaller ΔG_{H^*} value usually corresponds with the higher water dissociation energy barrier. Then, the high alkaline HER performance is supposed to correspond with the moderate ΔG_{H^*} value, not close to zero. In addition, the author mentioned ‘As illustrated in Fig.3g, the MoC-Mo₂C heterojunction exhibits better HER kinetics compared to those of bare MoC and bare Mo₂C.’ While, I cannot get any HER kinetics information from Fig. 3g. So, Fig.3 needs to be reorganized where some figures are not valuable anymore.

Response: Many thanks for the comments and suggestions. In our first-round calculations, we only discussed the ΔG_{H^*} value of bare MoC, bare Mo₂C and the MoC-Mo₂C heterojunction under acidic conditions. The corresponding results (in the original Fig. 3) exhibited that the ΔG_{H^*} value of the MoC-Mo₂C heterojunction in acid solution was more close to zero, which is consistent with our experimental results. For the HER in alkaline solution, we agree with the reviewer that the energy barriers of the two steps of water dissociation should be considered. Therefore, we did the calculation for reaction energy of water dissociation on MoC, Mo₂C, and MoC-Mo₂C including the two steps of water dissociation, and redraw the Fig. 3 that contained the calculated ΔG_{H^*} diagram of the HER in the acid solution at the equilibrium potential and the reaction energy diagram of water dissociation on MoC, Mo₂C, and MoC-Mo₂C in alkaline solution. It is obvious that the MoC-Mo₂C heterojunction exhibits better HER activity in both acid and alkaline solutions than that of bare MoC and Mo₂C, which is consistent with our experimental results.

Although the schematic illustration (Fig. 3g in the original manuscript, the Fig. 3d in the revised version) does not show any detailed reaction kinetics, we think it is useful for readers to understand the catalysis mechanism of the electrode. It directly indicates that the MoC-Mo₂C heterojunction exhibits a crucial role to the HER catalysis. Therefore, the schematic illustration (Fig. 3d in the new version) is kept in the revised manuscript. To avoid any confusion, we changed the expression (HER kinetics) to “HER activity”.

3. Finally, the calculated results by the software are just like the experimental measurement by the instrument. The authors need to give further theoretical analysis why the heterojunction structure exhibits high HER kinetics. Since structure determining the performance, what kind of intrinsic electronic structure corresponds with the heterojunction geometric structure and thus results in the high HER activity of Mo₂C-MoC, not Mo₂C or MoC. Having understanding this benefits the significance of this article and will be very helpful for readers to find more high-performance HER catalysts.

Response: Thank you very much for your comments and suggestions. We agree that the detailed analysis of both atomic and electronic structures of the heterojunction, especially the comparison with Mo₂C or MoC, will benefit the significance of the article. From the Bader charge transfer during the H adsorption in acid condition, we observed that the smaller the absolute value is, the corresponding ΔG_{H^*} value is closer to 0. And the DOS shows that the energy on MoC-Mo₂C is different for bare MoC or Mo₂C. It is probably that the combination of MoC and Mo₂C makes H desorption and adsorption easily, which is conducive to HER catalysis. The reaction paths in acidic and alkaline conditions are different so we divided the analysis into two parts. We have revised the theoretical analysis in the manuscript and updated Fig.3c, and Figure SI 25 accordingly.

In the revised manuscript, we have adjusted the expression of theoretical analysis. Page 13, light line 233: “The DFT calculations on both superficial and interfacial Mo sites of MoC, Mo₂C and MoC-Mo₂C were conducted to compare the electrocatalytic HER activity of bare MoC, Mo₂C and MoC-Mo₂C heterojunction. On acidic condition, among MoC (001), Mo₂C (101), and the MoC-Mo₂C, the MoC-Mo₂C interface exhibits the optimum Gibbs free energy of H* adsorption ($\Delta G_{H^*} = -0.13$ eV), as shown in Fig. 3b. The HER electrocatalyst with a positive value results in the poor adsorption of H*, while a catalyst with a negative value may lead to the difficult release of a H₂. The ideal value of $|\Delta G_{H^*}|$ should be zero. The smallest value of $|\Delta G_{H^*}|$ of the MoC-Mo₂C heterojunction indicates its better activity. The models of H* adsorption for Mo₂C (101) and MoC (001) are compared in Supplementary Fig. 21. The lattice mismatch of MoC and Mo₂C causes large local distortion at the heterostructure and changes the local electronic structure with more Mo³⁺/Mo²⁺ sites, which are consistent with our previous experiments. Bader charge analysis was conducted to analyze the charge transfer. H adatom adsorbed at the MoC-Mo₂C heterojunction possesses the least charge transfer of 0.35 e, corresponding to the smallest adsorption energy of -0.28 eV....”

Page 13, light line 261: “...showing that the MoC-Mo₂C is beneficial to activate hydrogen atoms. During the process of HER in alkaline solution, the first H₂O dissociates into intermediate H* and OH*. Then, the dissociation of the second H₂O leads to the generation of H₂. The HER energy diagrams were plotted to explain the

HER performance in alkaline solution. As shown in Fig. 3c, the highest energy barrier (the second H₂O dissociation step) of the MoC-Mo₂C heterojunction is 1.15 eV, while the HER performance on bare MoC and Mo₂C is relatively poor, with the highest energy barrier of 1.90 eV and 3.33 eV on bare MoC and Mo₂C, respectively. Overall, compared with bare MoC and Mo₂C, the MoC-Mo₂C heterojunction exhibits better HER activity in both acidic and alkaline solutions. As illustrated in Fig. 3d, the MoC-Mo₂C heterojunction exhibits better HER activity compared with those of bare MoC and bare Mo₂C....”

REVIEWER COMMENTS

Reviewer #4 (Remarks to the Author):

The authors did more detailed DFT calculations as my suggestions. They performed the calculation of the second water dissociation step. However, the authors still didn't completely answer my third question. They are supposed to explain why the energy barrier of water dissociation on the Mo₂C-MoC interface is lower than on Mo₂C or MoC. I still suggest the authors try to give more intrinsic theoretical explanations, not only simply listing the calculation results. In addition, the Figure R1 needs to give the relative energy values. Because there is only one water molecular in the first step but another water molecular is involved into the second step, so they don't meet the mass conservation in one potential energy surface. And what's more, I didn't find any detailed calculation results about the alkaline HER mechanism in supporting information section, such as the structure information of the reactant water adsorption, the dissociated intermediate, etc. They even didn't show where the active site is.

Detailed Responses to the Reviewers

Manuscript ID: NCOMMS-21-10682C

Title: A durable and pH-universal self-standing MoC-Mo₂C heterojunction electrode for efficient hydrogen evolution reaction

,

”

Author(s): Wei Liu^{1,2}, Xiting Wang³, Fan Wang^{1,2}, Kaifa Du^{1,2}, Zhaofu Zhang⁴, Yuzheng Guo^{3*}, Huayi Yin^{1,2*} and Dihua Wang^{1,2*}

We are grateful to the reviewers for their comments and suggestions that are helpful for us to improve our manuscript. We have revised the manuscript taking into full account of the reviewers' comments and recommendations. Below are our point-to-point responses in the same order as the reviewers' comments. The changes in the revised manuscript are highlighted in yellow.

Reviewer #4 (Remarks to the Author):

The authors did more detailed DFT calculations as my suggestions. They performed the calculation of the second water dissociation step. However, the authors still didn't completely answer my third question. They are supposed to explain why the energy barrier of water dissociation on the Mo₂C-MoC interface is lower than on Mo₂C or MoC. I still suggest the authors try to give more intrinsic theoretical explanations, not only simply listing the calculation results. In addition, the Figure R1 needs to give the relative energy values. Because there is only one water molecular in the first step but another water molecular is involved into the second step, so they don't meet the mass conservation in one potential energy surface. And what's more, I didn't find any detailed calculation results about the alkaline HER mechanism in supporting information section, such as the structure information of the reactant water adsorption, the dissociated intermediate, etc. They even didn't show where the active site is.

Response: We thank the referee for the valuable comments and suggestions. According to the comments and suggestions of the referee, our point-to-point responses are shown below.

Q1. They performed the calculation of the second water dissociation step. However, the authors still didn't completely answer my third question. They are supposed to explain why the energy barrier of water dissociation on the Mo₂C-MoC interface is lower than on Mo₂C or MoC. I still suggest the authors try to give more intrinsic theoretical explanations, not only simply listing the calculation results.

Response 1: We thank the referee for the valuable comments. We have added the following discussions in the main text to give more details about the mechanism of the water dissociation.

During the process of HER in alkaline solution, the first key step is that the first H₂O adsorbs on the surface and dissociates into intermediate H* and OH*. Then, the dissociation of the second H₂O leads to the generation of H₂ (Supplementary Fig. 26-28). The free energy diagrams are plotted to explain the corresponding HER performance in alkaline solution. As shown in Fig. 3c, the highest energy barrier (the

second H₂O dissociation step) of the MoC-Mo₂C heterojunction is 1.15 eV, while the highest energy barrier on bare MoC and Mo₂C are 1.90 and 3.33 eV, respectively. This indicates that the energy barrier of water dissociation on the MoC-Mo₂C interface is lower than that on MoC or Mo₂C, which can be explained by the interface Mo d orbital tuning since engineering the transition metal d orbitals is a feasible method to modulate the interaction between the molecule and active sites [Nature communications, 2019, 10(1), 1-8. and Advanced Materials, 2019, 31(16), 1807780.]. The occupied orbitals of the H₂O molecule are mainly p states. And the partial DOS shows that the electron transfer is mainly from the d orbitals of Mo atoms (Supplementary Fig. 29). The empty orbitals of MoC and Mo₂C near the Fermi level are more localized within the surface (Supplementary Fig. 30), suggesting that both MoC and Mo₂C are easy to capture the H₂O molecule but hard to dissociate the H₂O molecule. While the active Mo atoms on the MoC-Mo₂C interface have a higher partial DOS near Fermi level (Supplementary Fig. 29) and an empty hybridized d orbital perpendicular to the surface, which is not only beneficial to capture the H₂O molecule but also easy to dissociate the H₂O. In addition, the electron overlap between the active site and H₂O molecule on the MoC-Mo₂C interface is moderate compared to MoC and Mo₂C, which is helpful to decrease the energy barrier (Supplementary Fig. 31). Overall, the energy barrier of water dissociation on the MoC-Mo₂C interface is lower because of the Mo d orbital tuning. Therefore, the MoC-Mo₂C interface exhibits better HER activity than MoC and Mo₂C in both acid and alkaline solutions (Fig. 3d).

Supplementary Fig. 29 Partial density of states of Mo d states for MoC, Mo₂C, and MoC-Mo₂C.

Supplementary Fig. 30 Partial charge density for (a) MoC, (b) Mo₂C, and (c) MoC-Mo₂C heterojunction interface. The yellow region represents electron accumulation.

Supplementary Fig. 31 Partial charge density for (a) MoC, (b) Mo₂C, and (c) MoC-Mo₂C heterojunction interface with the adsorbed H₂O. The yellow region represents electron accumulation.

Q2. In addition, the Figure R1 needs to give the relative energy values. Because there is only one water molecular in the first step but another water molecular is involved into the second step, so they don't meet the mass conservation in one potential energy surface.

Response 2:

We thank the referee for the valuable comments and suggestions. According to the literatures [Nature communications, 2019, 10(1), 1-8. and Advanced Materials, 2019, 31(16), 1807780.], we have renamed y-axis to relative energy and modified the relative energy diagram. And the dash lines are added near the step of the second water molecular involved. We mainly focus on the energy barrier of H₂O dissociation in Fig 3c. The highest energy barrier (the second H₂O dissociation step) of the MoC-Mo₂C heterojunction is 1.15 eV, while the highest energy barrier on bare MoC and Mo₂C are 1.90 and 3.33 eV, respectively. This indicates that the energy barrier of water dissociation on the MoC-Mo₂C interface is lower than that on MoC or Mo₂C. The details are shown as follows:

Fig. 3 TOF LSV curves and DFT calculations. (a) TOF LSV curves of different electrodes. (b) Calculated ΔG_{H^*} diagram of the HER in acid electrolyte at the equilibrium potential. (c) Relative energy diagram of water dissociation on MoC, Mo₂C, and MoC-Mo₂C, including the two steps of water dissociation, in alkaline solution, TS: Transition State. (d) Schematic illustration of the HER mechanism.

Q3. And what's more, I didn't find any detailed calculation results about the alkaline HER mechanism in supporting information section, such as the structure information of the reactant water adsorption, the dissociated intermediate, etc. They even didn't show where the active site is.

Response 3: Thank you very much for your comments and suggestions. We have

given more details about the structure information of the reactant water adsorption, the dissociated intermediate on MoC, Mo₂C and MoC-Mo₂C interfaces, as shown in the new Figures S26, S27 and S28, respectively. And the charge density difference and partial charge density for MoC, Mo₂C and MoC-Mo₂C interfaces also have drawn in new Figure S30, which show the interaction difference between the active site and the surface effect.

Supplementary Fig. 26 The top-view and side-view structures of intermediates on the bare MoC during the process of alkaline HER.

Supplementary Fig. 27 The top-view and side-view structures of intermediates on the bare Mo₂C during the process of alkaline HER.

Supplementary Fig. 28 The top-view and side-view structures of intermediates on the MoC-Mo₂C during the process of alkaline HER.

Supplementary Fig. 30 Partial charge density for (a) MoC, (b) Mo₂C, and (c) MoC-Mo₂C heterojunction interface. The yellow region represents electron accumulation.

REVIEWERS' COMMENTS

Reviewer #4 (Remarks to the Author):

Comments have been addressed. I recommend it for publication.